# DP-REC: PRIVATE & COMMUNICATION-EFFICIENT FEDERATED LEARNING

## ABSTRACT

Privacy and communication efficiency are important challenges in federated training of neural networks, and combining them is still an open problem. In this work, we develop a method that unifies highly compressed communication and differential privacy (DP). We introduce a compression technique based on Relative Entropy Coding (REC) to the federated setting. With a minor modification to REC, we obtain a provably differentially private learning algorithm, DP-REC, and show how to compute its privacy guarantees. Our experiments demonstrate that DP-REC drastically reduces communication costs while providing privacy guarantees comparable to the state-of-the-art.

## 1 INTRODUCTION

The performance of modern neural-network-based machine learning models scales exceptionally well with the amount of data that they are trained on (Kaplan et al., 2020; Henighan et al., 2020). At the same time, industry (Xiao & Karlin), legislators (Dwork, 2019; Voigt & Von dem Bussche, 2017) and consumers (Laziuk, 2021) have become more conscious about the need to protect the privacy of the data that might be used in training such models. Federated learning (FL) describes a machine learning principle that enables learning on decentralized data by computing updates on-device. Instead of sending its data to a central location, a "client" in a federation of devices sends model updates computed on its data to the central server. Such an approach to learning from decentralized data promises to unlock the computing capabilities of billions of edge devices, enable personalized models and new applications in *e.g.* healthcare due to the inherently more private nature of the approach.

On the other hand, the federated paradigm brings challenges along many dimensions such as learning from non-i.i.d. data, resource-constrained devices, heterogeneous compute and communication capabilities, questions of fairness and representation, as well as the focus of our paper: communication overhead and characterization of privacy. Neural network training requires many passes over the data, resulting in repeated transfer of the model and updates between the server and the clients, potentially making communication a primary bottleneck (Kairouz et al., 2019; Wang et al., 2021). Compressing updates is an active area of research in FL and an essential step in "untethering" edge devices from WiFi. Moreover, while FL is intuitively more private through keeping data on-device, client updates have been shown to reveal sensitive information, even allowing to reconstruct a client's training data (Geiping et al., 2020). To truly protect client privacy, a more rigorous mathematical notion of Differential Privacy (DP) is widely adopted as the de-facto standard in FL. More specifically, DP for FL is usually defined at a client-level, which provides plausible deniability of an individual client's contribution to the federated model. Both aspects of FL, communication efficiency and differential privacy, have been extensively studied in separation. Effective combination of the two, however, is an open problem and an active area of research (Kairouz et al., 2021; Chen et al., 2020; Girgis et al., 2020). Some methods offer very limited compression capabilities for comparable utility (Kairouz et al., 2021), whereas others (Girgis et al., 2020) offer significant compression but require additional compromises, due to a looser privacy composition.

In this work, we present *Differentially Private Relative Entropy Coding* (`DP-REC`), a unified approach to jointly tackle privacy and communication efficiency. `DP-REC` makes use of the information-limiting constraints of DP to encode the client updates in FL with extremely short messages. First, we build our method based on compression without quantization; namely, the lossy variant of Relative Entropy Coding (REC), recently proposed by Flamich et al. (2020). Second, we show that it can be

modified to satisfy DP, and we provide a proof of its privacy guarantees along with the appropriate accounting technique. Third, we run extensive evaluation on 4 datasets and 3 types of models to demonstrate that our algorithm achieves extreme compression of client-to-server updates (down to 7 bits *per tensor*) at privacy levels $\varepsilon < 1$ with a small impact on utility ($< 6.5\%$ accuracy reduction on FEMNIST compared to `DP-FedAvg`). Additionally, we show how to reduce server-to-client communication by sending a history of updates accumulated since the client's last participation.

## 2 `DP-REC` FOR PRIVATE AND EFFICIENT COMMUNICATION IN FL

Federated learning has been described by McMahan et al. (2016) in the form of the `FedAvg` algorithm. At each communication round $t$, the server sends the current model parameters $\mathbf{w}^{(t)}$ to a subset $S^{(t)}$ of all clients $S$. Each chosen client $s$ updates the server-provided model $\mathbf{w}^{(t)}$, *e.g.*, via stochastic gradient descent, to better fit its local dataset $D_s = \{d_{si}\}_{i=1}^{N_s}$ with a given loss function

$$\mathcal{L}_s(\mathcal{D}_s, \mathbf{w}) := \frac{1}{N_s} \sum_{i=1}^{N_s} L(d_{si}, \mathbf{w}). \tag{1}$$

After $E$ epochs of optimization on the local dataset, the client-side optimization procedure results in an updated model $\mathbf{w}_s^{(t)}$, based on which the client computes its update to the global model

$$\mathbf{\Delta}_s^{(t)} = \mathbf{w}_s^{(t)} - \mathbf{w}^{(t)}; \tag{2}$$

and sends it to the server. The server then aggregates client-specific updates to get the new global model $\mathbf{w}^{(t+1)} = \mathbf{w}^{(t)} + \frac{1}{|S^{(t)}|} \sum_s \mathbf{\Delta}_s^{(t)} = \frac{1}{|S^{(t)}|} \sum_s \mathbf{w}_s^{(t)}$. Outside the context of differential privacy, client updates are usually weighted by the size $N_s$ of the local dataset. A generalization of this server-side scheme (Reddi et al., 2020) interprets $\frac{1}{|S^{(t)}|} \sum_{s \in S^{(t)}} \mathbf{\Delta}_s^{(t)}$ as a "gradient" for the server-side model and introduces more advanced updating schemes, such as Adam (Kingma & Ba, 2014).

Federated training involves repeated communication of model updates from clients to the server and vice versa. These updates can reveal sensitive information about the client data, so there is a need for formal privacy guarantees. The total communication cost can be significant, thus constraining FL to the use of unmetered channels, such as WiFi. Hence, compressing the update messages in a privacy-preserving way plays an important role in moving FL to a truly mobile use-case. In the following sections, we first describe the lossy version of Relative Entropy Coding (REC) (Flamich et al., 2020), then show how to extend it to the FL scenario before discussing it in the context of differential privacy. Finally, we show how `DP-REC` can be used to additionally compress server-to-client messages.

### 2.1 REC FOR EFFICIENT COMMUNICATION

Lossy REC, and its predecessor minimal random code learning (MIRACLE) (Havasi et al., 2018), have been originally proposed as a way to compress a random sample $\mathbf{w}$ from a distribution $q_\phi(\mathbf{w})$ parameterized with $\phi$, *i.e.*, $\mathbf{w} \sim q_\phi(\mathbf{w})$, by using information that is "shared" between the sender and the receiver. This information is given in terms of a shared prior distribution $p_\theta(\mathbf{w})$ with parameters $\boldsymbol{\theta}$ along with a shared random seed $R$. The sender proceeds by generating $K$ independent random samples, $\mathbf{w}_1, \ldots, \mathbf{w}_K$, from the prior distribution $p_\theta(\mathbf{w})$ according to the random seed $R$. Subsequently, it forms a categorical distribution $\tilde{q}_{\boldsymbol{\pi}}(\mathbf{w}_{1:K})$ over the $K$ samples with the probability of each sample being proportional to the likelihood ratio $\pi_k \propto q_\phi(\mathbf{w} = \mathbf{w}_k)/p_\theta(\mathbf{w} = \mathbf{w}_k)$. Finally, it draws a random sample $\mathbf{w}_{k^*}$ from $\tilde{q}_{\boldsymbol{\pi}}(\mathbf{w}_{1:K})$, corresponding to the $k^*$'th sample drawn from the shared prior. The sender can then communicate to the receiver the *index* $k^*$ with $\log_2 K$ bits. On the receiver side, $\mathbf{w}_{k^*}$ can be reconstructed by initializing the random number generator with $R$ and sampling the first $k^*$ samples from $p_\theta(\mathbf{w})$. These procedures are described in Algorithms 1 and 2.

Havasi et al. (2018) set $K$ to the exponential of the Kullback-Leibler (KL) divergence of $q_\phi(\mathbf{w})$ to the prior $p_\theta(\mathbf{w})$ with an extra constant $c$, *i.e.*, $K = \exp(\mathrm{KL}(q_\phi(\mathbf{w})\|p_\theta(\mathbf{w})) + c)$. In this case, the message length is at least $\mathcal{O}(\mathrm{KL}(q_\phi(\mathbf{w})\|p_\theta(\mathbf{w})))$. This can be theoretically motivated based on the work by Harsha et al. (2007); when the sender and the receiver share a source of randomness, under some assumptions, this KL divergence is a lower bound on the expected message length (Flamich et al., 2020). This brings forth an intuitive notion of compression that connects the compression rate

**Algorithm 1** The sender-side algorithm for lossy REC with the shared random seed $R$, the shared prior $p_{\boldsymbol{\theta}}(\mathbf{w})$ and the number of prior samples $K$.

Set state of pseudo-RNG to $R$
Draw $\mathbf{w}_1, \ldots, \mathbf{w}_K \overset{\text{i.i.d.}}{\sim} p_{\boldsymbol{\theta}}(\mathbf{w})$
$\alpha_k \leftarrow q_{\boldsymbol{\phi}}(\mathbf{w} = \mathbf{w}_k)/p_{\boldsymbol{\theta}}(\mathbf{w} = \mathbf{w}_k)$
$\pi_k \leftarrow \alpha_k / \sum_j \alpha_j$
$\mathbf{w}_{k^*} \sim \tilde{q}_{\boldsymbol{\pi}}(\mathbf{w}_{1:K})$
**return** $k^*$ encoded with $\log_2 K$ bits

**Algorithm 2** The receiver-side algorithm for lossy REC.

Receive $k^*$ from the sender
Set state of pseudo-RNG to $R$
Draw $\mathbf{w}_1, \ldots, \mathbf{w}_{k^*} \overset{\text{i.i.d.}}{\sim} p_{\boldsymbol{\theta}}(\mathbf{w})$
Use $\mathbf{w}_{k^*}$ for downstream task

with the amount of additional information encoded in $q_{\boldsymbol{\phi}}(\mathbf{w})$ relative to the information in $p_{\boldsymbol{\theta}}(\mathbf{w})$; the smaller the amount of extra information the shorter the message length will be and, in the extreme case where $q_{\boldsymbol{\phi}}(\mathbf{w}) = p_{\boldsymbol{\theta}}(\mathbf{w})$, the message length will be $\mathcal{O}(1)$. Of course, achieving this efficiency is meaningless if the bias of this procedure is high; fortunately, Havasi et al. (2018) show that for appropriate values of $c$ and under mild assumptions, the bias, namely $|\mathbb{E}_{\tilde{q}_{\boldsymbol{\pi}}(\mathbf{w}_{1:K})}[f] - \mathbb{E}_{q_{\boldsymbol{\phi}}(\mathbf{w})}[f]|$ for arbitrary functions $f$, can be sufficiently small. In all of our subsequent discussions, we parametrize $K$ as a function of a binary bit-width $b$, *i.e.*, $K = 2^b$, and treat $b$ as the hyperparameter.

## 2.2 REC FOR EFFICIENT COMMUNICATION IN FL

We can adapt this procedure to FL by appropriately choosing distributions over client-to-server messages (model updates), $q_{\boldsymbol{\phi}_s}^{(t)}(\boldsymbol{\Delta}_s^{(t)})$, along with the prior distribution $p_{\boldsymbol{\theta}}^{(t)}(\boldsymbol{\Delta}_s^{(t)})$ on each round $t$:

$$p_{\boldsymbol{\theta}}^{(t)}(\boldsymbol{\Delta}_s^{(t)}) := \mathcal{N}(\boldsymbol{\Delta}_s^{(t)}|\mathbf{0}, \sigma^2\mathbf{I}), \qquad q_{\boldsymbol{\phi}_s}^{(t)}(\boldsymbol{\Delta}_s^{(t)}) := \mathcal{N}(\boldsymbol{\Delta}_s^{(t)}|\mathbf{w}_s^{(t)} - \mathbf{w}^{(t)}, \sigma^2\mathbf{I}), \qquad (3)$$

*i.e.*, for the prior we use a Gaussian distribution centered at zero with appropriately chosen $\sigma$ and for the message distribution we opt for a Gaussian with the same standard deviation centered at the model update. The form of $q$ is chosen to provide a plug-in solution to potentially resource constrained devices, as well as to readily satisfy the differential privacy constrains discussed in Section 2.3. Note that, as opposed to the `FedAvg` client update definition in (2), here we consider $\boldsymbol{\Delta}_s^{(t)}$ to be a random variable and the difference $\mathbf{w}_s^{(t)} - \mathbf{w}^{(t)}$ to be the mean of the client-update distribution $q$ over $\boldsymbol{\Delta}_s^{(t)}$.

Let us now see how communication efficiency is realized in the FL pipeline. The length of the message will be a function of how much "extra" information about the local dataset $D_s$ is encoded in $\mathbf{w}_s^{(t)}$, measured by KL divergence. As we show later, this has a nice interplay with DP: the DP constraints bound the amount of information encoded in each update, resulting in highly compressible messages. It is also worth noting that this procedure can be done parameter-wise (*i.e.*, communicate $\log_2 K$ bits per parameter), layer-wise ($\log_2 K$ bits for each layer in the network) or even network-wise ($\log_2 K$ bits in total). Any arbitrary intermediate vector size is also possible. This is done by splitting $\boldsymbol{\Delta}_s^{(t)}$ into $M$ independent groups (which is possible due to our assumption of factorial distributions over the dimensions of the vector) and applying compression independently to each group (see also Theorem 3). If we have many groups (*e.g.*, when we perform per-parameter compression), we can boost the compression even further by entropy coding the indices with their empirical distribution.

## 2.3 PRIVATE & EFFICIENT COMMUNICATION FOR FL WITH DP-REC

In order to make the compression procedure described in Section 2.2 differentially private, we need to bound the sensitivity of the mechanism and quantify the noise inherent to it. Similarly to `DP-FedAvg`, bounding the sensitivity consists of clipping the norm of client updates $\mathbf{w}_s^{(t)} - \mathbf{w}^{(t)}$. In the context of REC, this means that the client message distribution $q_{\boldsymbol{\phi}}^{(t)}$ cannot be too different from the server prior $p_{\boldsymbol{\theta}}^{(t)}$ in any given round $t$. After this step, instead of explicitly injecting additional noise to the updates, we make use of the fact that the procedure in itself is stochastic. Two sources of randomness play a role in each round $t$: (1) drawing $K$ samples from the prior $p_{\boldsymbol{\theta}}^{(t)}$; (2) drawing an update from the importance sampling distribution $\tilde{q}_{\boldsymbol{\pi}}^{(t)}$. We coin the name Differentially-Private Relative Entropy Coding (`DP-REC`) for the resulting mechanism, outlined in Algorithms 3 and 4.

**Algorithm 3** The client-side algorithm for DP-REC at a given round $t$. $R_s^{(t)}$ is the client-chosen shared random seed, $\mathbf{w}^{(t)}$ is the server model, $\mathbf{w}_s^{(t)}$ is the fine-tuned model of client $s$, $K$ is the number of prior samples, $C$ is the clipping threshold and $\sigma$ is the prior standard deviation.

---

$R_s^{(t)} \leftarrow$ choose and set client-specific seed for round $t$
$\boldsymbol{\phi}_s^{(t)} \leftarrow \mathbf{w}_s^{(t)} - \mathbf{w}^{(t)}$
$\boldsymbol{\Delta}_1, \ldots, \boldsymbol{\Delta}_K \overset{\text{i.i.d.}}{\sim} \mathcal{N}(\boldsymbol{\Delta}_s^{(t)}|\mathbf{0}, \sigma^2\mathbf{I})$
$\hat{\boldsymbol{\phi}}_s^{(t)} = \boldsymbol{\phi}_s^{(t)} \min(1, C/\|\boldsymbol{\phi}_s^{(t)}\|_2)$
$\alpha_k \leftarrow \mathcal{N}(\boldsymbol{\Delta}_s^{(t)} = \boldsymbol{\Delta}_k|\hat{\boldsymbol{\phi}}_s^{(t)}, \sigma^2\mathbf{I})/\mathcal{N}(\boldsymbol{\Delta}_s^{(t)} = \boldsymbol{\Delta}_k|\mathbf{0}, \sigma^2\mathbf{I})$
$\pi_k \leftarrow \alpha_k/\sum_j \alpha_j$
$\boldsymbol{\Delta}_{k^*} \sim \tilde{q}_{\boldsymbol{\pi}}^{(t)}(\boldsymbol{\Delta}_{1:K})$
**return** $k^*$ encoded with $\log_2 K$ bits, $R_s^{(t)}$

---

**Algorithm 4** The server side algorithm for DP-REC.

---

Receive $k^*, R_s^{(t)}$ from client $s$
Set state of pseudo-RNG to $R_s^{(t)}$
$\boldsymbol{\Delta}_1, \ldots, \boldsymbol{\Delta}_{k^*} \overset{\text{i.i.d.}}{\sim} \mathcal{N}(\boldsymbol{\Delta}_s^{(t)}|\mathbf{0}, \sigma^2\mathbf{I})$
Use $\boldsymbol{\Delta}_{k^*}$ for downstream task

---

Finally, to prove that DP-REC is differentially private and calculate the corresponding $\varepsilon, \delta$, we build upon prior work in privacy accounting for ML and FL; particularly, on Rényi differential privacy (RDP) (Mironov, 2017) and the moments accountant (Abadi et al., 2016). Our analysis in the next section follows the established approach based on tail bounds of the privacy loss random variable, addressing such important differences as two sources of randomness and non-Gaussianity of $\tilde{q}_{\boldsymbol{\pi}}^{(t)}$. Additional discussion on other aspects of our method can be found in Appendices B.1 and D.2.

## 2.4 PRIVACY ANALYSIS OF DP-REC

A possible avenue to analyze privacy for DP-REC is to derive DP bounds relying on Theorem 3.2 of Havasi et al. (2018). However, obtaining reasonable guarantees can be challenging, as the probability bound in (Havasi et al., 2018, Theorem 3.2) has to be incorporated in $\delta$ and it is at least $e^{-0.125 \ln K}$. As such, a fairly common value $\delta = 10^{-5}$ would require $\sim e^{92}$ samples from the prior. To overcome this issue, we consider an importance sampling bound by Agapiou et al. (2017, Section 2.2). It scales (inversely) linearly with the number of samples, helping for smaller sample sizes.

Let us restate the bound for completeness. For some test function $\zeta$, one can characterize the bound between the true expectation and the importance sampling estimate in the following way:

$$\sup_{|\zeta|<1} \left|\mathbb{E}\left[\mu^N(\zeta) - \mu(\zeta)\right]\right| \leq \frac{12}{K}e^{\mathcal{D}_2(q_\phi||p_\theta)}, \tag{4}$$

where $K$ is the number of samples, $\mu^N(\cdot) = \mathbb{E}_{\tilde{q}_{\boldsymbol{\pi}}}[\cdot]$, $\mu(\cdot) = \mathbb{E}_{q_\phi}[\cdot]$ and $\mathcal{D}_2(\cdot||\cdot)$ denotes the Rényi divergence of order 2 between the target $q_\phi$ and the proposal $p_\theta$. Our choice of $\zeta$ (*cf.* Appendix A.1) guarantees $|\zeta| < 1$ for any input and enables the use of this bound.

Theorem 1 summarizes our main theoretical result. It enables DP-REC to combine the server-side privacy accounting in the central model of DP with extreme model update compression, while allowing clients to privatize their updates locally and protect against an honest-but-curious server.

**Theorem 1.** *After $T$ rounds, with the client-to-server bitrate $b$, DP-REC is differentially private with*

$$\delta \leq \min_\lambda e^{-\lambda\varepsilon} \prod_{t=1}^{T} e^{(\lambda-1)c_\lambda^{(t)} + \lambda c_{\lambda+1}^{(t)}} + \frac{12}{2^b}\sum_{t=1}^{T} e^{c_2^{(t)}},$$

*where the constant $c_\lambda^{(t)} \geq \mathcal{D}_\lambda\left(q_\phi^{(t)}||p_\theta^{(t)}\right)$ is the upper bound on the Rényi divergence of order $\lambda$ between the client model update distribution $q_\phi^{(t)}$ and the server prior $p_\theta^{(t)}$ in any given round $t$.*

*Proof.* See Appendix A.1. □

An attentive reader may notice a strong resemblance between Theorem 1 and the moments accountant (Abadi et al., 2016) and Rényi DP (Mironov, 2017) results. Indeed, it turns out that the DP-REC

compression procedure yields the privacy loss that can be bound by almost the double of the normal continuous Gaussian mechanism loss, plus a small overhead, and composed in a very similar manner.

**Shared random seed** Prior work relying on shared random seeds has been shown to be vulnerable to attacks due to the possibility of "inverting" the noise (Kairouz et al., 2021). `DP-REC` is not susceptible to this for two reasons. First, the randomness of the method comes from two sources instead of one, and knowing one seed does not permit to fully reconstruct the un-noised sample. In our guarantee, $\delta$ corresponds to the probability of the mechanism failure w.r.t. both sources of randomness, making it as secure as the standard Gaussian mechanism even when one of the seeds is shared. However, this alone is not sufficient because the shared seed $R$ could be manipulated to generate a set of samples on which to encode the update such that the guarantee would not hold (*e.g.*, in the tails of the privacy loss distribution). Hence, we add the second line of defense as we shift the task of picking $R$ to a client, who then transmits the seed along with the index $k^*$ for decoding. This way, neither the server nor any other entity can manipulate the seed to break the privacy guarantee.

**Privacy amplification** Like many recent approaches, including (Kairouz et al., 2021), `DP-REC` can be seen as a hybrid method, privatizing updates locally and providing an amplified central guarantee. A key difference, however, is that our method does not calibrate noise to the aggregate. Namely, the scale of noise and the guarantee do not depend on whether a thousand, a hundred, or just one client is sampled in a round. This is due to non-additive nature of our mechanism, which does not let us easily benefit from the variance reduction as the Gaussian mechanism does. As a result, we always calibrate to an "aggregate" of one client. While this property increases the necessary noise, it has an upside of allowing stronger privacy amplification. Substituting the typical sampling *without replacement* by sampling *with replacement*, we can use the amplification factor of $1/|S|$ while accounting for $T|S'|$ rounds, instead of $|S'|/|S|$ while accounting for $T$ rounds, and obtain a tighter overall guarantee.

**Secure aggregation** One of the considerations in FL is that the server might be honest-but-curious rather than fully trusted. In `DP-FedAvg` (McMahan et al., 2017), the noise addition is delegated to the server, and thus, secure aggregation (Bonawitz et al., 2017) is necessary to prevent the server from receiving client updates in the clear. `DDGauss` (Kairouz et al., 2021) and analogous methods mitigate this problem by shifting the noise addition to the client side, but still use secure aggregation for the reasons stated in the previous paragraph: noise is calibrated to the aggregate and may be insufficient to protect some clients. In `DP-REC`, because of the noise calibration to individual updates, we believe that the provided local guarantee (which can be computed by removing subsampling amplification) is sufficient against non-malicious servers. To further boost privacy protection, an important future work direction is integrating the `DP-REC` compression scheme with secure aggregation.

**Interplay of privacy and compression** Theorem 1 relates privacy to our compression scheme in an intuitive manner. In order to get meaningful privacy guarantees, we have to bound the Rényi divergence in any given round $t$ for any $\lambda \geq 1$, which limits the amount of information encoded in $q_\phi^{(t)}$ relative to $p_\theta^{(t)}$. As a result, enforcing DP guarantees also implies that our scheme will have highly compressible messages. We formalize this interplay in the following lemma.

**Lemma 2.** *Consider compressing the expected value of $\zeta$ under the $q_{\phi_s}(\boldsymbol{\Delta}_s)$ at (3), and let $\xi$ be a desired average compression bias of REC for $\zeta$. To achieve this target, a sufficient bitrate $b$ is at most*

$$b \leq \log_2 12 + \frac{1}{\log 2} \mathcal{D}_2 \left( q_{\phi_s} || p_\theta \right) - \log_2 \frac{\xi}{G},$$

*where $|\zeta| \leq G$ and $\mathcal{D}_2 \left( q_{\phi_s} || p_\theta \right)$, and thus the maximum bitrate $b$, is controlled by the chosen $(\varepsilon, \delta)$.*

*Proof.* The proof follows from (4), by setting $K = 2^b$, rearranging the terms and adjusting for $G$. $\square$

We also verify the claim empirically (Figure 2), fixing the bound on the Rényi divergence and showing that the performance with extreme compression (*i.e.*, 7 bits/tensor) and no compression is similar. Finally, to confirm that various granularity of compression (per-parameter, per-layer, per-network, etc.) does not interfere with our privacy analysis, we introduce and prove the following theorem.

**Theorem 3** (Compression-by-parts). *Expectation of an arbitrary function $\zeta(\boldsymbol{\Delta}^{[1]}, \ldots, \boldsymbol{\Delta}^{[M]})$ over the importance sampling distributions $q_{\boldsymbol{\pi}}^{[1]}, \ldots, q_{\boldsymbol{\pi}}^{[M]}$, built for $M$ non-intersecting parameter groups*

*independently, is equivalent to the expectation over the joint distribution $q_{\boldsymbol{\pi}}$ built on $2^{\sum_{i=1}^{M} b_i}$ samples:*

$$\mathbb{E}_{\boldsymbol{\Delta}^{[1]} \sim q_{\boldsymbol{\pi}}^{[1]}} \left[ \ldots \mathbb{E}_{\boldsymbol{\Delta}^{[M]} \sim q_{\boldsymbol{\pi}}^{[M]}} \left[ \zeta(\boldsymbol{\Delta}^{[1]}, \ldots, \boldsymbol{\Delta}^{[M]}) \right] \ldots \right] = \mathbb{E}_{\boldsymbol{\Delta}^{[1:M]} \sim q_{\boldsymbol{\pi}}} \left[ \zeta(\boldsymbol{\Delta}^{[1:M]}) \right].$$

*Proof.* See Appendix A.2. □

### 2.5 Compressing server-to-client messages

The compression procedure described in Section 2.2 is a specific example of (stochastic) vector quantization where the shared codebook is determined by a shared random seed. Here we show how the principle of communicating indices into such a shared codebook additionally allows for the compression of the server-to-client communication. Instead of sending the full server-side model to a specific client, the server can choose to collect all updates to the global model in-between two subsequent rounds that the client participates in. Based on this history of codebook indices, the client can deterministically reconstruct the current state of the server model before beginning local optimization. Clearly, the expected length of the history is proportional to the total number of clients and the amount of client subsampling we perform during training. At the beginning of a round, the server can therefore compare the bit-size of the history and choose to send the full-precision model $\mathbf{w}^{(t)}$ instead. Taking a model with $1k$ parameters as an example, a single uncompressed model update is approximately equal to $4k$ communicated indices when using 8-bit codebook compression of the whole model. Crucially, compressing server-to-client messages this way has no influence on the DP nature of `DP-REC` since the local model of DP ensures privacy of client information (and the stronger central guarantee applies as long as the secrecy of the sample is preserved within the updates history).

For clients participating in their first round of training, the first seed without accompanying indices can be understood as seeding the random initialization of the server-side model. Algorithms 5 and 6 (in Appendix B.2) describe this scheme. It is important to note that the client-side update rule must be equal to the server-side update rule, *i.e.* in generalized `FedAvg` (Reddi et al., 2020) it might be necessary to additionally send the optimizer state when sending the current global model $\mathbf{w}^{(t)}$.

## 3 Related work

The privacy promise of federated learning relies heavily on the use of additional techniques, such as differential privacy and secure multi-party computation, to provide rigorous theoretical guarantees. Without these techniques, FL has been shown to be vulnerable to attacks (Geiping et al., 2020) and unintended leakage of sensitive information (Thakkar et al., 2020). One of the main open challenges is to reduce the communication cost while preserving DP guarantees (Kairouz et al., 2019).

McMahan et al. (2017) outlined `DP-FedAvg`, which has since been the staple of DP federated learning. In this default scheme, clients clip norms of their updates before submitting them to the server (preferably, using a secure aggregation protocol (Bonawitz et al., 2017)). The server then completes the DP mechanism by adding Gaussian noise to the aggregate of multiple clients. Without secure aggregation, this method allows clients to compress their updates and reduce communication, but becomes vulnerable to a malicious or honest-but-curious server. One could use the local model of DP (Dwork et al., 2014) instead of the central model to address this issue: clients would add noise locally before sending the updates. But this leads to a pronounced drop in accuracy due to the larger scale of noise necessary in the local model (Kairouz et al., 2019). The few existing examples that use the local model operate on very large numbers of clients and updates (Pihur et al., 2018).

To overcome this problem of excessive noise, researchers are increasingly looking in the direction of hybrid solutions (Shi et al., 2011; Rastogi & Nath, 2010; Agarwal et al., 2018; Truex et al., 2019). The key idea of these techniques is to reduce the variance of the locally added noise by taking into account the larger global number of clients and accounting DP in the central model. Since the smaller noise variance is insufficient to protect individual updates, secure aggregation is necessary for these methods. In this context, the local noise distributions have to be discrete, such that their sum after secure aggregation is also discrete and is of known shape, allowing the server to compute guarantees centrally. As a result, until recently, most of these methods relied on the binomial distribution. Compared to the traditional Gaussian mechanisms, it is at a disadvantage because it does not yield Renyi or concentrated DP, and thus cannot benefit from tighter adaptive composition and

amplification through sampling, which is particularly important in ML applications (Kairouz et al., 2021). Kairouz et al. (2021) presented a novel analysis of the discrete Gaussian mechanism (Canonne et al., 2020) in terms of concentrated DP. Their mechanism, named Discrete Distributed Gaussian (`DDGauss`), is adapted to the context of FL, can provide accuracy comparable to the centralized Gaussian mechanism, and enables parameter-wise quantization. However, at higher compression rates (*e.g.*, 12 bits per parameter), `DDGauss` fails to train a model for reasonable $\varepsilon$ values.

Finally, there is recent work featuring extreme compression rates for DP algorithms, bearing a resemblance to our solution. Girgis et al. (2020) proposed to use a locally private mechanism along with a secure shuffler to communicate models one (privatized) parameter at a time. It compresses client messages to $\log d$ bits for a model of $d$ parameters. This approach, however, seems to require a large number of clients to participate in a single round (10k for their MNIST experiment), which is impractical in realistic scenarios and detrimental to the total communication cost (upload + download). Chen et al. (2020) achieve similarly high compression rates as (Girgis et al., 2020); however, they do not consider the context of federated learning but rather focus on frequency and mean estimation. As we did not have an FL baseline for (Chen et al., 2020) and could not reliably reproduce the results of Girgis et al. (2020), we focused our comparison with these methods on mean estimation. Details can be found in Appendix D.2. In short, we find that the method of Chen et al. (2020) has an edge over `DP-REC` if a good prior is lacking, making it preferable for low-information, "one-shot" scenarios, such as mean estimation. On the other hand, with a good prior (or one that is learned during training), `DP-REC` performs better and thus is more appropriate in federated settings.

## 4 EXPERIMENTS

We evaluate `DP-REC` on several non-i.i.d. FL tasks that provide client-level privacy guarantees: image classification on a non-i.i.d. split of MNIST (LeCun et al., 1998) into 100 clients and FEM-NIST (Caldas et al., 2018) into 3.5k clients, along with next character prediction on the Shakespeare dataset (Caldas et al., 2018) with LSTMs (Hochreiter & Schmidhuber, 1997) and 660 clients as well as tag prediction on the StackOverflow (SO-LR) dataset (TFF Authors, 2019) with a logistic regression model (Kairouz et al., 2021) and 342477 clients. For baselines, we consider `DP-FedAvg` as the gold standard without compression and `DDGauss` as a baseline that involves parameter quantization. Both of these methods target central DP guarantees and employ secure aggregation. All exeriments were implemented in PyTorch (Paszke et al., 2019). We provide experimental details in Appendix C.

### 4.1 RESULTS AND DISCUSSION

We plot the global model accuracy for different privacy budgets $\varepsilon$ as a function of the total communication cost in Figure 1; this highlights how efficiently each method spends bits of communication in order to reach a target accuracy. Due to the substantial compression by `DP-REC` relative to the baselines, we use *log-scale* for the x-axis of communication costs. Additionally, Figure 3 in Appendix D shows the accuracy achieved as a function of the privacy budget $\varepsilon$; this highlights how efficiently each method spends its privacy budget in order to reach a specific target accuracy. Note that in certain cases the training is stopped before convergence, due to exhausted privacy budget.

In Table 1, we report the final model performance and total communication costs for different privacy budgets. It should be mentioned that the evaluation loop for StackOverflow is prohibitively expensive to be done in each round due to having ~1.6M datapoints. We thus pick 10k random datapoints for evaluation during training to plot the learning curves (not necessarily the same for each run), following (Reddi et al., 2020). The numbers in Table 1 refer to the model performance at the end of training, where we evaluate on the entire test set for `DP-FedAvg` and `DP-REC`. For `DDGauss`, since Kairouz et al. (2021) do not report the model performance on the full test set, we use the numbers shown at the end of their plot. Finally, results for some settings are omitted because of either (i) convergence issues for stricter privacy budgets (a typical phenomenon for $\varepsilon < 1$ on smaller federations); (ii) not being reported by the related work we compare with; or (iii) not being necessary in light of sufficient performance under better privacy guarantees (e.g., on StackOverflow).

Observing the experimental results, we clearly see the trade-offs. `DP-REC` can drastically reduce the *total communication costs* (download and upload) of federated training depending on the subsampling rate and amount of clients in a federation. For certain cases, we can get extreme reduction, such

Table 1: Performance ($\pm$ standard error obtained via multiple runs) and total communication (in GB), achieved for different privacy guarantees $\varepsilon$. Results for `DDGauss` (marked `DDGx` for $x$ bits) are taken from (Kairouz et al., 2021), which uses a slightly smaller version of the FEMNIST dataset ($3.4k$ clients instead of $3.5k$).

| **MNIST** | `DP-FedAvg` | `DP-REC` |
|---|---|---|
| Acc. ($\varepsilon=3$) | $82.1 \pm 0.8$ | $66.5 \pm 1.5$ |
| Acc. ($\varepsilon=6$) | $90.0 \pm 0.4$ | $79.0 \pm 1.9$ |
| *Comm.* | *43* | *0.01* |

| **Shakespeare** | `DP-FedAvg` | `DP-REC` |
|---|---|---|
| Acc. ($\varepsilon=3$) | $39.0 \pm 0.1$ | $29.0 \pm 0.1$ |
| *Comm.* | *81* | *0.1* |

| **FEMNIST** | `DP-FedAvg` | `DDG16` | `DDG12` | `DP-REC` |
|---|---|---|---|---|
| Acc. ($\varepsilon=1$) | $65.7 \pm 0.3$ | $-$ | $-$ | $59.3 \pm 0.1$ |
| Acc. ($\varepsilon=3$) | $74.2 \pm 0.1$ | ~72 | ~25 | $67.0 \pm 0.1$ |
| Acc. ($\varepsilon=6$) | $75.5 \pm 0.2$ | ~77 | ~71 | $69.1 \pm 0.1$ |
| *Comm.* | *259* | *194* | *178* | *14.2* |

| **SO LR** | `DP-FedAvg` | `DDG16` | `DDG12` | `DP-REC` |
|---|---|---|---|---|
| R@5 ($\varepsilon=1$) | $19.3 \pm 0.0$ | ~14 | ~10 | $18.4 \pm 0.7$ |
| *Comm.* | *3356* | *2517* | *2307* | *32* |

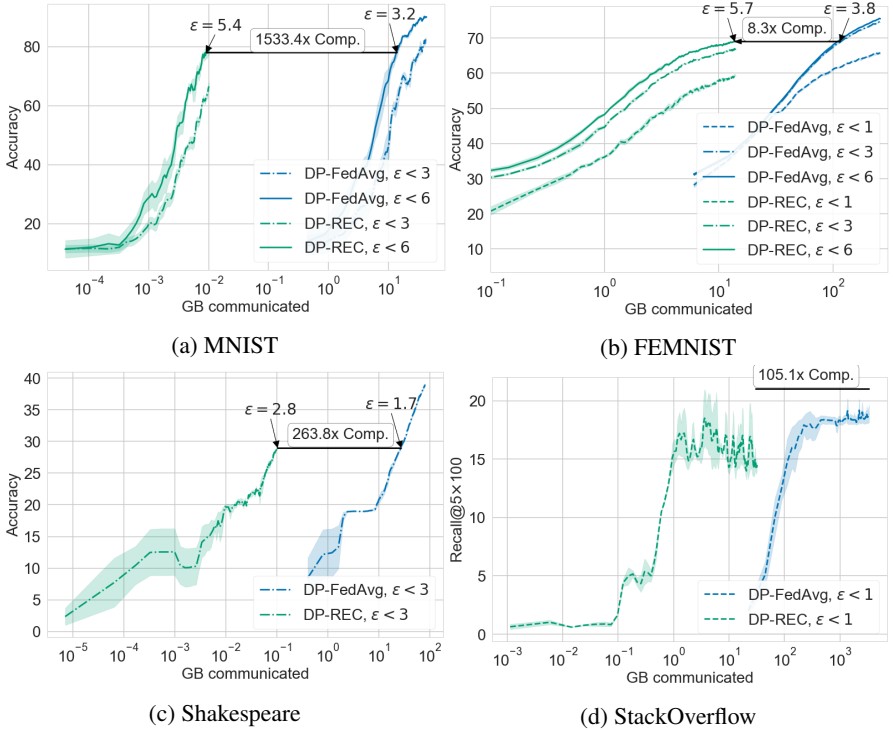

(a) MNIST      (b) FEMNIST

(c) Shakespeare      (d) StackOverflow

Figure 1: Test accuracy (%) with $\pm 1$ standard error as a function of communication (in GB, in *log-scale*). For StackOverflow, we report compression at the end of training, as the learning curves are based on different subsets of the validation set.

as MNIST with 4300x compression. But even for other cases, the reduction is still several orders of magnitude (105x for SO-LR and 18x on FEMNIST). Nevertheless, `DP-FedAvg` and `DDGauss` (depending on the task) can reach better model performance for a given privacy budget. This is primarily due to two reasons. Firstly, both `DP-FedAvg` and `DDGauss` add noise calibrated to the number of clients in a given round to get a central DP guarantee on the aggregate; in contrast, `DP-REC` calibrates noise to the contribution of individual updates (see Section 2.4). The signal-to-noise-ratio for the model updates is thus worse for `DP-REC`. Secondly, in `DP-REC` we have to clip client updates more aggressively to account for the privacy loss overhead incurred from the REC compression (roughly double for each iteration compared to Gaussian mechanism). This can be mitigated in larger federations, since the clipping can be reduced based on stronger privacy amplification for the central model of DP. For example, observe the StackOverflow experiment, where the accuracy delta between `DP-REC` and `DP-FedAvg` is smaller. Furthermore, it is precisely in those cases that reducing the communication cost is important from a practical perspective. When we target a specific accuracy

within the reach of both `DP-REC` and `DP-FedAvg`, we compress between 8.3x (FEMNIST) to 1533x (MNIST), at the additional cost of 1.9 to 2.2 in $\varepsilon$, relative to `DP-FedAvg`.

**Comparison with local DP guarantees**  The nature of the `DP-REC` privacy mechanism requires calibrating noise to individual client updates rather than their contribution to the aggregate. This property warrants a different kind of comparison: equalizing local guarantees for each individual update (*i.e.*, as seen when the secrecy of the sample in client sampling is not preserved). We use our non-i.i.d. FEMNIST setting with 1.5k rounds and tune the noise of `DP-FedAvg` such that a single client update, without any privacy amplification, has the same $\varepsilon_{\text{local}}$ guarantee. When training `DP-FedAvg` with a local-DP guarantee that `DP-REC` obtains when targeting central-DP with $\varepsilon = 3$, we see that `DP-FedAvg` obtains 68.9% accuracy, whereas `DP-REC` gets 63.4%. On a setting where we target the local-DP guarantee that `DP-REC` gets when aiming for central-DP with $\varepsilon = 1$, `DP-FedAvg` achieves 60.1% accuracy compared to `DP-REC`'s 57.4%. In both cases, `DP-REC` achieves an overall compression rate of 72.5x with 7-bits per tensor, while losing 2.7% to 5.5% in accuracy compared to `DP-FedAvg` (depending on the privacy target) due to more aggressive clipping. It is also worth noting that the performance delta relative to the results shown in Table 1 is smaller, highlighting the negative effects of calibrating the noise to an "aggregate" of a single client.

**Compressibility of differentially private updates**  As we noted in Section 2.4, differential privacy has a positive interplay with the `REC` compression scheme. DP upper-bounds the Rényi divergence for any order $\lambda$ between the proposal distribution $p_{\boldsymbol{\theta}}^{(t)}$ and the model update distribution $q_{\boldsymbol{\phi}}^{(t)}$ and, according to lemma 2, this limits the maximum bitrate $b$ of information to be communicated. In order to empirically verify this claim, we consider the non-i.i.d. FEMNIST classification with `DP-REC`, and hyperparameters that target a model with $\varepsilon = 3$ after 1.5k rounds. We then maintain the same bound on Rényi divergence and vary the number of bits per tensor from 4 to 7. We also include a variant without compression, which directly communicates a random sample from the clipped update distribution $q_{\boldsymbol{\phi}}^{(t)}$. The results appear to verify our claim and can be seen in Figure 2. After 4 bits, we see very small improvements with adding additional bits and, with 7 bits, the performance is very similar to the uncompressed baseline.

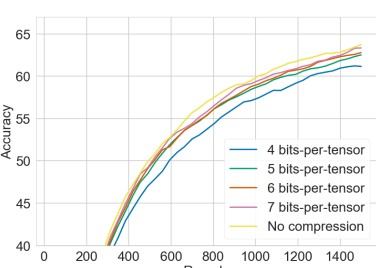

Figure 2: Effect of the bit-width on model performance with `DP-REC` for clipping targeting $\varepsilon = 3$ after 1.5k rounds. The "no-compression" variant communicates a random sample directly from $q_{\boldsymbol{\phi}}^{(t)}$ instead of compressing it via `DP-REC`.

## 5  CONCLUSION

With `DP-REC`, we formalized the intuitive notion that $\varepsilon, \delta$ differentially private messages necessarily contain a small amount of information and can therefore be compressed significantly. A bound on the Rényi divergence between the server-side prior $p$ and the locally optimized $q$ implies a small message size for `REC`, elegantly tying together DP and communication efficiency for federated learning. The nature of our bound in Theorem 1 reveals the flipside of `DP-REC`. As opposed to the standard Gaussian mechanism in central DP, it requires stronger clipping, effectively reducing the utility of models trained with `DP-REC` for a given privacy budget. Our experiments with the StackOverflow dataset show that these limitations can be mitigated by the stronger privacy amplification in situations with large numbers of clients. Contrary to intuition, spending more bits to communicate updates from clients to the server cannot recover this utility as the necessary information is lost in clipping, *i.e.*, before forming the importance sampling distribution $\tilde{q}$.

There are several important directions left for future work. The first would be to investigate the convergence of `DP-REC`. Intuitively, we can expect the algorithm to be convergent if `DP-FedAvg` is convergent. The main difference of `DP-REC` is that the gradient is sampled using importance weights rather than the true Gaussian probabilities. Asymptotically, these two distributions should be close, and the bias of the compressed gradients would be bounded by a small value (as discussed in Section 2.4). Empirically, we observe no issues with convergence. The second would be to investigate secure shuffling, similarly to Girgis et al. (2020) as a way to amplify our privacy further; this can lead to less aggressive clipping and thus improved performance for a given privacy budget.

ETHICS STATEMENT

This paper addresses a significant, present-day problem and provides a clear, detailed explanation of our solution to facilitate reproducibility and promote research integrity. We expect our work to have an overall beneficial societal impact through its main contributions of communication reduction and differential privacy. One potential concern worth mentioning is the interplay between differential privacy and fairness in FL, an actively researched open problem. Bagdasaryan et al. (2019) demonstrate that DP training can have a disproportionate effect on underrepresented and more complex sub-populations, resulting in a disparate accuracy reduction. At the same time, Hooker et al. (2020) showed that compression of models can have negative impacts by amplifying biases. In the context of FL, compression is performed on model updates instead of a final model, therefore requiring further research on its influence on bias amplification.

REPRODUCIBILITY STATEMENT

For the theoretical results, we clearly state our assumptions and present complete proofs in the appendix. For experimental evaluation, although we cannot provide the source code at the time of the submission because it is proprietary, we include extensive information on algorithms and settings in the appendix, and repeat our experiments with multiple random seeds to aid reproducibility.

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

APPENDIX

# A    PROOFS

## A.1    PROOF OF THEOREM 1

**Theorem 1.** *After $T$ rounds, with the client-to-server bitrate $b$,* `DP-REC` *is differentially private with*

$$\delta \leq \min_\lambda e^{-\lambda \varepsilon} \prod_{t=1}^{T} e^{(\lambda-1)c_\lambda^{(t)} + \lambda c_{\lambda+1}^{(t)}} + \frac{12}{2^b} \sum_{t=1}^{T} e^{c_2^{(t)}},$$

*where the constant $c_\lambda^{(t)} \geq \mathcal{D}_\lambda \left( q_\phi^{(t)} || p_\theta^{(t)} \right)$ is the upper bound on the Rényi divergence of order $\lambda$ between the client model update distribution $q_\phi^{(t)}$ and the server prior $p_\theta^{(t)}$ in any given round $t$.*

*Proof.* Consider a privacy mechanism $\mathcal{A} : \mathbb{D} \to \mathbb{R}^d$, mapping a dataset $D \in \mathbb{D}$ to a $d$-dimensional model update $\Delta \in \mathbb{R}^d$. Recall that our mechanism features two sources of randomness: drawing from distributions $p_\theta$ and then $\tilde{q}_\pi$ (which is based on $q_\phi$ and $p_\theta$).

Similarly to the derivations for the moments accountant (Abadi et al., 2016), we can write

$$\Pr[\mathcal{A}(D) \in \Omega] = \Pr[\mathcal{A}(D) \in \Omega \cap \bar{\mathcal{S}}] + \Pr[\mathcal{A}(D) \in \Omega \cap \mathcal{S}] \tag{5}$$

$$\leq e^\varepsilon \Pr[\mathcal{A}(D') \in \Omega] + \Pr[\mathcal{A}(D) \in \mathcal{S}], \tag{6}$$

where $\Omega \subset \mathbb{R}^d$ is an arbitrary set of outcomes, $\mathcal{S} = \{\Delta_{k^*} : L_{\tilde{q}_\pi}(\Delta_{k^*}, D, D') > \varepsilon\}$ is the set of outcomes where the bound on privacy loss is violated, and $\bar{\mathcal{S}}$ denotes a complement. For multiple iterations, $k^*$ can be viewed as the multi-index picked from the importance sampling distribution across all iterations.

First, let us make a brief side-note on the privacy loss. As we pursue the goal of client-level privacy, we consider two clients with different local datasets $D$ and $D'$. Their update distributions are parameterized by $\phi$ for one client and $\phi'$ for another: $q_\phi^{(1:T)}(\cdot)$ and $q_{\phi'}^{(1:T)}(\cdot)$. We will denote the corresponding importance sampling distributions as $\tilde{q}_\pi^{(1:T)}(\cdot)$ and $\tilde{q}_{\pi'}^{(1:T)}(\cdot)$. Then the privacy loss for these two clients is

$$L_{\tilde{q}_\pi^{(1:T)} \| \tilde{q}_{\pi'}^{(1:T)}} \left( \Delta_{k^*}^{(1:T)}, D, D' \right) = \log \frac{\tilde{q}_\pi^{(1:T)} \left( \Delta_{k^*}^{(1:T)} \right)}{\tilde{q}_{\pi'}^{(1:T)} \left( \Delta_{k^*}^{(1:T)} \right)} \tag{7}$$

$$= \log \frac{\tilde{q}_\pi^{(1:T)}(\cdot)}{\tilde{p}_\pi^{(1:T)}(\cdot)} + \log \frac{\tilde{q}_{\pi'}^{(1:T)}(\cdot)}{\tilde{p}_\pi^{(1:T)}(\cdot)} \tag{8}$$

$$= \log \frac{\tilde{q}_\pi^{(1:T)}(\cdot)}{\tilde{p}_\pi^{(1:T)}(\cdot)} + \log \frac{\tilde{q}_{\pi'}^{(1:T)}(\cdot)}{\tilde{p}_{\pi'}^{(1:T)}(\cdot)} \tag{9}$$

$$= L_{\tilde{q}_\pi^{(1:T)}}(\cdot) + L_{\tilde{q}_{\pi'}^{(1:T)}}(\cdot), \tag{10}$$

where $\tilde{p}_\pi^{(1:T)} = \tilde{p}_{\pi'}^{(1:T)}$ is the uniform distribution over $K$ samples from $p_\theta^{(1:T)}$. Thus, it is sufficient to bound the privacy loss $L_{\tilde{q}_\pi^{(1:T)}}(\cdot)$ for the worst-case $\phi$ with some $\varepsilon, \delta$. For the centralized guarantee, it would correspond to bounding the influence of adding or removing one client. If the local guarantee, or bounding the influence of substituting a client, is necessary, it will be given by $2\varepsilon$ (and correspondingly, $2\delta$). This is consistent with prior work where bounds on substitution are also double the bounds on addition/removal.

Consider the second term of Eq. 6 for $T$ rounds of training:

$$\Pr\left[L_{\tilde{q}_{\pi}^{(1:T)}} > \varepsilon\right] \tag{11}$$

$$= \int p_{\boldsymbol{\theta}}^{(1:T)}\left(\boldsymbol{\Delta}_{1:K}^{(1)} \cdots \boldsymbol{\Delta}_{1:K}^{(T)}\right) \quad \sum_{k_1=1}^{K} \tilde{q}_{\boldsymbol{\pi}}^{(1)}\left(\boldsymbol{\Delta}_{k_1}^{(1)}\right) \cdots$$

$$\sum_{k_T=1}^{K} \tilde{q}_{\boldsymbol{\pi}}^{(T)}\left(\boldsymbol{\Delta}_{k_T}^{(T)}\middle|\boldsymbol{\Delta}_{1:K}^{(1:T-1)}\right) \mathbb{1}_{\{L_{\tilde{q}_{\boldsymbol{\pi}}} > \varepsilon\}} d\boldsymbol{\Delta}_{1:K}^{(1:T)}, \tag{12}$$

where

$$L_{\tilde{q}_{\boldsymbol{\pi}}}^{(1:T)} = \log \frac{\tilde{q}_{\boldsymbol{\pi}}\left(\boldsymbol{\Delta}_{1:K}^{(1:T)}\right)}{\tilde{p}_{\boldsymbol{\pi}}\left(\boldsymbol{\Delta}_{1:K}^{(1:T)}\right)} \tag{13}$$

is the total privacy loss across all samples from all iterations. Note also that by $\tilde{p}_{\boldsymbol{\pi}}(\cdot)$ we denote an importance sampling distribution formed when the proposal and the target are the same, which is essentially uniform over the $K$ samples.

Since $\mathbb{1}_{\{\cdot\}} \leq 1$, we can employ the bound (4) directly. However, due to the iterative importance sampling over rounds and possible dependencies between rounds, we have to use the law of total expectation and apply the bound recursively. Namely, denoting $\zeta^{(T)}(\boldsymbol{\Delta}_{1:K}^{(1:T)}) = \mathbb{1}_{\{L_{\tilde{q}_{\boldsymbol{\pi}}} > \varepsilon\}}$,

$$\Pr[L_{\tilde{q}_{\boldsymbol{\pi}}^{(1:T)}} > \varepsilon]$$

$$= \int p_{\boldsymbol{\theta}}^{(1)}\left(\boldsymbol{\Delta}_{1:K}^{(1)}\right) \sum_{k_1=1}^{K} \tilde{q}_{\boldsymbol{\pi}}^{(1)}\left(\boldsymbol{\Delta}_{k_1}^{(1)}\right) \cdots \tag{14}$$

$$\int p_{\boldsymbol{\theta}}^{(T)}\left(\boldsymbol{\Delta}_{1:K}^{(T)}\middle|\boldsymbol{\Delta}_{1:K}^{(1:T-1)}\right) \sum_{k_T=1}^{K} \tilde{q}_{\boldsymbol{\pi}}^{(T)}\left(\boldsymbol{\Delta}_{k_T}^{(T)}\middle|\boldsymbol{\Delta}_{1:K}^{(1:T-1)}\right) \zeta^{(T)}\left(\boldsymbol{\Delta}_{1:K}^{(1:T)}\right) d\boldsymbol{\Delta}_{1:K}^{(1:T)},$$

$$\leq \int p_{\boldsymbol{\theta}}^{(1)}\left(\boldsymbol{\Delta}_{1:K}^{(1)}\right) \sum_{k_1=1}^{K} \tilde{q}_{\boldsymbol{\pi}}^{(1)}\left(\boldsymbol{\Delta}_{k_1}^{(1)}\right) \cdots \tag{15}$$

$$\int p_{\boldsymbol{\theta}}^{(T)}\left(\boldsymbol{\Delta}_{1:K}^{(T)}\middle|\boldsymbol{\Delta}_{1:K}^{(1:T-1)}\right)\left[\int q_{\boldsymbol{\phi}}^{(T)}\left(\boldsymbol{\Delta}^{(T)}\middle|\boldsymbol{\Delta}_{1:K}^{(1:T-1)}\right) \zeta^{(T)}\left(\boldsymbol{\Delta}_{1:K}^{(1:T)}\right) d\boldsymbol{\Delta}^{(T)} + \frac{12}{K}\rho_T\right] d\boldsymbol{\Delta}_{1:K}^{(1:T)}$$

$$\leq \int p_{\boldsymbol{\theta}}^{(1)}\left(\boldsymbol{\Delta}_{1:K}^{(1)}\right) \sum_{k_1=1}^{K} \tilde{q}_{\boldsymbol{\pi}}^{(1)}\left(\boldsymbol{\Delta}_{k_1}^{(1)}\right) \cdots \tag{16}$$

$$\underbrace{\int p_{\boldsymbol{\theta}}^{(T)}\left(\boldsymbol{\Delta}_{1:K}^{(T)}\middle|\boldsymbol{\Delta}_{1:K}^{(1:T-1)}\right) \int q_{\boldsymbol{\phi}}^{(T)}\left(\boldsymbol{\Delta}^{(T)}\middle|\boldsymbol{\Delta}_{1:K}^{(1:T-1)}\right) \zeta^{(T)}\left(\boldsymbol{\Delta}_{1:K}^{(1:T)}\right) d\boldsymbol{\Delta}^{(T)} d\boldsymbol{\Delta}_{1:K}^{(T)} d\boldsymbol{\Delta}_{1:K}^{(1:T-1)}}_{\zeta^{(T-1)}\left(\boldsymbol{\Delta}_{1:K}^{(1:T-1)}\right)}$$

$$+ \frac{12}{K}\rho_T, \tag{17}$$

where we can pull (17) out due to its independence from the rest of the expectation.

As noted in line (16), one can then treat the inner expectations (over the distributions from round $T$) as a new function $\zeta^{(T-1)}(\boldsymbol{\Delta}_{1:K}^{(1:T-1)})$, which is still bounded by 1 because it's an expectation of an indicator function. Repeating the procedure, we get

$$\Pr[L_{\tilde{q}_{\boldsymbol{\pi}}^{(1:T)}} > \varepsilon]$$

$$\leq \mathbb{E}_{p_{\boldsymbol{\theta}}^{(1:T)}}\left[\mathbb{E}_{q_{\boldsymbol{\phi}}^{(1:T)}}\left[\mathbb{1}_{\left\{\log \frac{\tilde{q}_{\boldsymbol{\pi}}\left(\boldsymbol{\Delta}_{1:K}^{(1:T)}\right)}{\tilde{p}_{\boldsymbol{\pi}}\left(\boldsymbol{\Delta}_{1:K}^{(1:T)}\right)} > \varepsilon\right\}}\right]\right] + \underbrace{\frac{12}{2^b}\sum_{t=1}^{T}\rho_t}_{\rho^{(1:T)}}, \tag{18}$$

with $\rho_t = e^{\mathcal{D}_2\left(q_\phi^{(t)} \| p_\theta^{(t)}\right)}$. It is worth noting one more time that $q_\phi^{(t)}$ and $p_\theta^{(t)}$ are conditional distributions, depending on rounds $1...t-1$, and that we do not assume independence between rounds.

Applying Chernoff inequality to (18):

$$\Pr[L_{\tilde{q}_\pi^{(1:T)}} > \varepsilon]$$

$$\leq e^{-\lambda\varepsilon} \mathbb{E}_{p_\theta^{(1:T)}} \left[ \mathbb{E}_{q_\phi^{(1:T)}} \left[ e^{\lambda \log \frac{\tilde{q}_\pi\left(\boldsymbol{\Delta}_{1:K}^{(1:T)}\right)}{\tilde{p}_\pi\left(\boldsymbol{\Delta}_{1:K}^{(1:T)}\right)}} \right] \right] + \frac{12}{2^b} \rho^{(1:T)} \tag{19}$$

$$= e^{-\lambda\varepsilon} \mathbb{E}_{p_\theta^{(1:T)}} \left[ \mathbb{E}_{q_\phi^{(1:T)}} \left[ \left( \frac{\tilde{q}_\pi\left(\boldsymbol{\Delta}_{1:K}^{(1:T)}\right)}{\tilde{p}_\pi\left(\boldsymbol{\Delta}_{1:K}^{(1:T)}\right)} \right)^\lambda \right] \right] + \frac{12}{2^b} \rho^{(1:T)}. \tag{20}$$

As we have done above in (14), we re-arrange expectations using the law of total expectation:

$$= e^{-\lambda\varepsilon} \mathbb{E}_{p_\theta^{(1)}} \left[ \mathbb{E}_{q_\phi^{(1)}} \left[ \ldots \mathbb{E}_{p_\theta^{(T)}} \left[ \mathbb{E}_{q_\phi^{(T)}} \left[ \left( \frac{\tilde{q}_\pi\left(\boldsymbol{\Delta}_{1:K}^{(1:T)}\right)}{\tilde{p}_\pi\left(\boldsymbol{\Delta}_{1:K}^{(1:T)}\right)} \right)^\lambda \right] \right] \ldots \right] \right] + \frac{12}{2^b} \rho^{(1:T)}. \tag{21}$$

Analogously, let us apply the chain rule to the inner expression:

$$\left( \frac{\tilde{q}_\pi\left(\boldsymbol{\Delta}_{k^*}^{(1:T)}\right)}{\tilde{p}_\pi\left(\boldsymbol{\Delta}_{k^*}^{(1:T)}\right)} \right)^\lambda = \underbrace{\left( \frac{\tilde{q}_\pi\left(\boldsymbol{\Delta}_{k^*}^{(1)}\right)}{\tilde{p}_\pi\left(\boldsymbol{\Delta}_{k^*}^{(1)}\right)} \right)^\lambda}_{\ell_{\tilde{q}_\pi^{(1)}}} \cdot \ldots \cdot \underbrace{\left( \frac{\tilde{q}_\pi\left(\boldsymbol{\Delta}_{k^*}^{(T)} \middle| \boldsymbol{\Delta}_{k^*}^{(1:T-1)}\right)}{\tilde{p}_\pi\left(\boldsymbol{\Delta}_{k^*}^{(T)} \middle| \boldsymbol{\Delta}_{k^*}^{(1:T-1)}\right)} \right)^\lambda}_{\ell_{\tilde{q}_\pi^{(T)}}} \tag{22}$$

Continuing on (18):

$$= e^{-\lambda\varepsilon} \mathbb{E}_{p_\theta^{(1)}} \left[ \mathbb{E}_{q_\phi^{(1)}} \left[ \ldots \mathbb{E}_{p_\theta^{(T)}} \left[ \mathbb{E}_{q_\phi^{(T)}} \left[ \ell_{\tilde{q}_\pi^{(1)}} \cdot \ldots \cdot \ell_{\tilde{q}_\pi^{(T)}} \right] \right] \ldots \right] \right] + \frac{12}{2^b} \rho^{(1:T)} \tag{23}$$

$$= e^{-\lambda\varepsilon} \mathbb{E}_{p_\theta^{(1)}} \left[ \mathbb{E}_{q_\phi^{(1)}} \left[ \ell_{\tilde{q}_\pi^{(1)}} \ldots \underbrace{\mathbb{E}_{p_\theta^{(T)}} \left[ \mathbb{E}_{q_\phi^{(T)}} \left[ \ell_{\tilde{q}_\pi^{(T)}} \right] \right]}_{L_{\tilde{q}_\pi}^{(T)} \leq \kappa_\lambda} \ldots \right] \right] + \frac{12}{2^b} \rho^{(1:T)}. \tag{24}$$

If we bound the quantity $L_{\tilde{q}_\pi}^{(T)}$ by some constant $\kappa_\lambda$, independent of all the previous samples $\boldsymbol{\Delta}_{k^*}^{(1:T-1)}$, we can bring it in front of the rest of the expectation. Note that this quantity is not exactly the privacy loss of the mechanism in round $T$, and the slight abuse of notation is for simplicity purposes.

By again performing this operation recursively,

$$\leq e^{-\lambda\varepsilon} \kappa_\lambda^T + \frac{12}{2^b} \rho^{(1:T)}. \tag{25}$$

Let us therefore consider any of such terms in isolation. To proceed, observe that we can switch from importance sampling to the original continuous distributions inside the expectation in the following way:

$$\frac{\tilde{q}_\pi(\boldsymbol{\Delta}_{k^*}^{(t)})}{\tilde{p}_\pi(\boldsymbol{\Delta}_{k^*}^{(t)})} = \frac{q_\phi(\boldsymbol{\Delta}_{k^*}^{(t)})/p_\theta(\boldsymbol{\Delta}_{k^*}^{(t)})}{p_\theta(\boldsymbol{\Delta}_{k^*}^{(t)})/p_\theta(\boldsymbol{\Delta}_{k^*}^{(t)})} \frac{\sum_k p_\theta(\boldsymbol{\Delta}_k^{(t)})/p_\theta(\boldsymbol{\Delta}_k^{(t)})}{\sum_k q_\phi(\boldsymbol{\Delta}_k^{(t)})/p_\theta(\boldsymbol{\Delta}_k^{(t)})}$$

$$= \frac{q_\phi(\boldsymbol{\Delta}_{k^*}^{(t)})}{p_\theta(\boldsymbol{\Delta}_{k^*}^{(t)})} \frac{1}{\sum_k q_\phi(\boldsymbol{\Delta}_k^{(t)})/p_\theta(\boldsymbol{\Delta}_k^{(t)})} \sum_k \frac{p_\theta(\boldsymbol{\Delta}_k^{(t)})}{q_\phi(\boldsymbol{\Delta}_k^{(t)})} \frac{q_\phi(\boldsymbol{\Delta}_k^{(t)})}{p_\theta(\boldsymbol{\Delta}_k^{(t)})}$$

$$= \frac{q_\phi(\boldsymbol{\Delta}_{k^*}^{(t)})}{p_\theta(\boldsymbol{\Delta}_{k^*}^{(t)})} \mathbb{E}_{\boldsymbol{\Delta}_{k'}^{(t)} \sim \tilde{q}_\pi} \left[ \frac{p_\theta(\boldsymbol{\Delta}_{k'}^{(t)})}{q_\phi(\boldsymbol{\Delta}_{k'}^{(t)})} \right] \tag{26}$$

Hence, keeping in mind that expectations and distributions are conditioned on the previous $t-1$ rounds,

$$L_{\tilde{q}_{\boldsymbol{\pi}}}^{(t)} = \mathbb{E}_{p_{\boldsymbol{\theta}}^{(t)}} \left[ \mathbb{E}_{q_{\boldsymbol{\phi}}^{(t)}} \left[ e^{\lambda \log \frac{q_{\boldsymbol{\phi}}\left(\boldsymbol{\Delta}^{(t)}\right)}{p_{\boldsymbol{\theta}}\left(\boldsymbol{\Delta}^{(t)}\right)} \mathbb{E}_{\tilde{q}_{\boldsymbol{\pi}}^{(t)}} \left[ \frac{p_{\boldsymbol{\theta}}\left(\boldsymbol{\Delta}_{k'}^{(t)}\right)}{q_{\boldsymbol{\phi}}\left(\boldsymbol{\Delta}_{k'}^{(t)}\right)} \right]} \right] \right] \tag{27}$$

$$= \mathbb{E}_{q_{\boldsymbol{\phi}}^{(t)}} \left[ e^{\lambda \log \frac{q_{\boldsymbol{\phi}}\left(\boldsymbol{\Delta}^{(t)}\right)}{p_{\boldsymbol{\theta}}\left(\boldsymbol{\Delta}^{(t)}\right)}} \right] \mathbb{E}_{p_{\boldsymbol{\theta}}^{(t)}} \left[ e^{\lambda \log \mathbb{E}_{\tilde{q}_{\boldsymbol{\pi}}^{(t)}} \left[ \frac{p_{\boldsymbol{\theta}}\left(\boldsymbol{\Delta}_{k'}^{(t)}\right)}{q_{\boldsymbol{\phi}}\left(\boldsymbol{\Delta}_{k'}^{(t)}\right)} \right]} \right] \tag{28}$$

$$= \mathbb{E}_{q_{\boldsymbol{\phi}}^{(t)}} \left[ \left( \frac{q_{\boldsymbol{\phi}}\left(\boldsymbol{\Delta}^{(t)}\right)}{p_{\boldsymbol{\theta}}\left(\boldsymbol{\Delta}^{(t)}\right)} \right)^{\lambda} \right] \mathbb{E}_{p_{\boldsymbol{\theta}}^{(t)}} \left[ \mathbb{E}_{\tilde{q}_{\boldsymbol{\pi}}^{(t)}} \left[ \frac{p_{\boldsymbol{\theta}}\left(\boldsymbol{\Delta}_{k'}^{(t)}\right)}{q_{\boldsymbol{\phi}}\left(\boldsymbol{\Delta}_{k'}^{(t)}\right)} \right]^{\lambda} \right]. \tag{29}$$

The first expectation is equivalent to the one in DP-SGD and is basically a moment-generating function of the privacy loss random variable between two sequences of Gaussian distributions (or mixtures when subsampling is used) over $T$ rounds.

Let us consider the second expectation, which requires further manipulation to avoid using the privacy sensitive importance weights. We cannot employ the bound by Agapiou et al. (2017) because the

function inside not bounded. However, we can utilize the special form of this function:

$$
\mathbb{E}_{p_{\boldsymbol{\theta}}^{(t)}} \left[ \mathbb{E}_{\tilde{q}_{\boldsymbol{\pi}}^{(t)}} \left[ \frac{p_{\boldsymbol{\theta}}\left(\boldsymbol{\Delta}_{k'}^{(t)}\right)}{q_{\boldsymbol{\phi}}\left(\boldsymbol{\Delta}_{k'}^{(t)}\right)} \right]^{\lambda} \right]
\tag{30}
$$

$$
= \mathbb{E}_{\boldsymbol{\Delta}_{1:K}^{(t)} \sim p_{\boldsymbol{\theta}}^{(t)}} \left[ \frac{\sum_l q_{\boldsymbol{\phi}}(\boldsymbol{\Delta}_l^{(t)})/p_{\boldsymbol{\theta}}(\boldsymbol{\Delta}_l^{(t)})}{K} \frac{K}{\sum_l q_{\boldsymbol{\phi}}(\boldsymbol{\Delta}_l^{(t)})/p_{\boldsymbol{\theta}}(\boldsymbol{\Delta}_l^{(t)})} \mathbb{E}_{\tilde{q}_{\boldsymbol{\pi}}^{(t)}} \left[ \frac{p_{\boldsymbol{\theta}}\left(\boldsymbol{\Delta}_{k'}^{(t)}\right)}{q_{\boldsymbol{\phi}}\left(\boldsymbol{\Delta}_{k'}^{(t)}\right)} \right]^{\lambda} \right]
\tag{31}
$$

$$
= \mathbb{E}_{\boldsymbol{\Delta}_{1:K}^{(t)} \sim p_{\boldsymbol{\theta}}^{(t)}} \left[ \frac{\sum_l q_{\boldsymbol{\phi}}(\boldsymbol{\Delta}_l^{(t)})/p_{\boldsymbol{\theta}}(\boldsymbol{\Delta}_l^{(t)})}{K} \sum_k \frac{\frac{q_{\boldsymbol{\phi}}(\boldsymbol{\Delta}_k^{(t)})}{p_{\boldsymbol{\theta}}(\boldsymbol{\Delta}_k^{(t)})} \frac{p_{\boldsymbol{\theta}}(\boldsymbol{\Delta}_k^{(t)})}{q_{\boldsymbol{\phi}}(\boldsymbol{\Delta}_k^{(t)})}}{\sum_l q_{\boldsymbol{\phi}}(\boldsymbol{\Delta}_l^{(t)})/p_{\boldsymbol{\theta}}(\boldsymbol{\Delta}_l^{(t)})} \mathbb{E}_{\tilde{q}_{\boldsymbol{\pi}}^{(t)}} \left[ \frac{p_{\boldsymbol{\theta}}\left(\boldsymbol{\Delta}_{k'}^{(t)}\right)}{q_{\boldsymbol{\phi}}\left(\boldsymbol{\Delta}_{k'}^{(t)}\right)} \right]^{\lambda} \right]
\tag{32}
$$

$$
= \mathbb{E}_{\boldsymbol{\Delta}_{1:K}^{(t)} \sim p_{\boldsymbol{\theta}}^{(t)}} \left[ \frac{\sum_l q_{\boldsymbol{\phi}}(\boldsymbol{\Delta}_l^{(t)})/p_{\boldsymbol{\theta}}(\boldsymbol{\Delta}_l^{(t)})}{K} \mathbb{E}_{\tilde{q}_{\boldsymbol{\pi}}^{(t)}} \left[ \frac{p_{\boldsymbol{\theta}}\left(\boldsymbol{\Delta}_{k'}^{(t)}\right)}{q_{\boldsymbol{\phi}}\left(\boldsymbol{\Delta}_{k'}^{(t)}\right)} \right]^{\lambda+1} \right]
\tag{33}
$$

$$
\leq \mathbb{E}_{\boldsymbol{\Delta}_{1:K}^{(t)} \sim p_{\boldsymbol{\theta}}^{(t)}} \left[ \frac{\sum_l q_{\boldsymbol{\phi}}(\boldsymbol{\Delta}_l^{(t)})/p_{\boldsymbol{\theta}}(\boldsymbol{\Delta}_l^{(t)})}{K} \mathbb{E}_{\tilde{q}_{\boldsymbol{\pi}}^{(t)}} \left[ \left( \frac{p_{\boldsymbol{\theta}}\left(\boldsymbol{\Delta}_{k'}^{(t)}\right)}{q_{\boldsymbol{\phi}}\left(\boldsymbol{\Delta}_{k'}^{(t)}\right)} \right)^{\lambda+1} \right] \right]
\tag{34}
$$

$$
= \mathbb{E}_{\boldsymbol{\Delta}_{1:K}^{(t)} \sim p_{\boldsymbol{\theta}}^{(t)}} \left[ \frac{\sum_l q_{\boldsymbol{\phi}}(\boldsymbol{\Delta}_l^{(t)})/p_{\boldsymbol{\theta}}(\boldsymbol{\Delta}_l^{(t)})}{K} \sum_k \frac{q_{\boldsymbol{\phi}}(\boldsymbol{\Delta}_k^{(t)})/p_{\boldsymbol{\theta}}(\boldsymbol{\Delta}_k^{(t)})}{\sum_l q_{\boldsymbol{\phi}}(\boldsymbol{\Delta}_l^{(t)})/p_{\boldsymbol{\theta}}(\boldsymbol{\Delta}_l^{(t)})} \left( \frac{p_{\boldsymbol{\theta}}(\boldsymbol{\Delta}_k^{(t)})}{q_{\boldsymbol{\phi}}(\boldsymbol{\Delta}_k^{(t)})} \right)^{\lambda+1} \right]
\tag{35}
$$

$$
= \mathbb{E}_{\boldsymbol{\Delta}_{1:K}^{(t)} \sim p_{\boldsymbol{\theta}}^{(t)}} \left[ \frac{1}{K} \sum_k \frac{q_{\boldsymbol{\phi}}(\boldsymbol{\Delta}_k^{(t)})}{p_{\boldsymbol{\theta}}(\boldsymbol{\Delta}_k^{(t)})} \left( \frac{p_{\boldsymbol{\theta}}(\boldsymbol{\Delta}_k^{(t)})}{q_{\boldsymbol{\phi}}(\boldsymbol{\Delta}_k^{(t)})} \right)^{\lambda+1} \right]
\tag{36}
$$

$$
= \frac{1}{K} \sum_k \mathbb{E}_{\boldsymbol{\Delta}_{1:K}^{(t)} \sim p_{\boldsymbol{\theta}}^{(t)}} \left[ \frac{q_{\boldsymbol{\phi}}(\boldsymbol{\Delta}_k^{(t)})}{p_{\boldsymbol{\theta}}(\boldsymbol{\Delta}_k^{(t)})} \left( \frac{p_{\boldsymbol{\theta}}(\boldsymbol{\Delta}_k^{(t)})}{q_{\boldsymbol{\phi}}(\boldsymbol{\Delta}_k^{(t)})} \right)^{\lambda+1} \right]
\tag{37}
$$

$$
= \frac{1}{K} \sum_k \mathbb{E}_{\boldsymbol{\Delta}_k^{(t)} \sim q_{\boldsymbol{\phi}}} \left[ \left( \frac{p_{\boldsymbol{\theta}}(\boldsymbol{\Delta}_k^{(t)})}{q_{\boldsymbol{\phi}}(\boldsymbol{\Delta}_k^{(t)})} \right)^{\lambda+1} \right]
\tag{38}
$$

$$
= \mathbb{E}_{\boldsymbol{\Delta}^{(t)} \sim q_{\boldsymbol{\phi}}} \left[ \left( \frac{p_{\boldsymbol{\theta}}(\boldsymbol{\Delta}^{(t)})}{q_{\boldsymbol{\phi}}(\boldsymbol{\Delta}^{(t)})} \right)^{\lambda+1} \right]
\tag{39}
$$

$$
= e^{\lambda \mathcal{D}_{\lambda+1}(p_{\boldsymbol{\theta}} || q_{\boldsymbol{\phi}})}
\tag{40}
$$

Putting everything together, we have

$$
L_{\tilde{q}_{\boldsymbol{\pi}}}^{(t)} \leq e^{(\lambda-1)\mathcal{D}_{\lambda}(q_{\boldsymbol{\phi}} || p_{\boldsymbol{\theta}}) + \lambda \mathcal{D}_{\lambda+1}(p_{\boldsymbol{\theta}} || q_{\boldsymbol{\phi}})} \leq \kappa_{\lambda},
\tag{41}
$$

where $\kappa_{\lambda}$ does not depend on any of the previous rounds, since we can bound Rényi divergences (e.g., by clipping the model updates), and is defined as

$$
\kappa_{\lambda} \equiv \max_{\boldsymbol{\phi}, \boldsymbol{\theta}} e^{(\lambda-1)\mathcal{D}_{\lambda}(q_{\boldsymbol{\phi}} || p_{\boldsymbol{\theta}}) + \lambda \mathcal{D}_{\lambda+1}(p_{\boldsymbol{\theta}} || q_{\boldsymbol{\phi}})}.
\tag{42}
$$

It then satisfies the conditions to obtain Eq. 25, and consequently, proves the theorem with

$$
c_{\lambda}^{(t)} = \max \begin{cases} \max_{\boldsymbol{\theta}, \boldsymbol{\phi}} \mathcal{D}_{\lambda}(q_{\boldsymbol{\phi}}^{(t)} || p_{\boldsymbol{\theta}}^{(t)}) \\ \max_{\boldsymbol{\theta}, \boldsymbol{\phi}} \mathcal{D}_{\lambda}(p_{\boldsymbol{\theta}}^{(t)} || q_{\boldsymbol{\phi}}^{(t)}) \end{cases}
\tag{43}
$$

$\square$

## A.2 COMPRESSION BY PARTS

Let us consider the setting where parts of the model (such as tensors, or even individual parameters) are compressed independently. Assume all model parameters $\boldsymbol{\Delta}$ are split into $M$ non-intersecting groups $\boldsymbol{\Delta}^{[1]}, \ldots, \boldsymbol{\Delta}^{[M]}$ and are encoded using $b_1, \ldots, b_M$ bits correspondingly. We use square brackets to distinguish these indices from the round indices. The following holds.

**Theorem 3.** *Expectation of an arbitrary function $\zeta(\boldsymbol{\Delta}^{[1]}, \ldots, \boldsymbol{\Delta}^{[M]})$ over the importance sampling distributions $q_{\boldsymbol{\pi}}^{[1]}, \ldots, q_{\boldsymbol{\pi}}^{[M]}$, built according to the outlined procedure for each parameter group independently, is equivalent to the expectation over the joint importance sampling distribution $q_{\boldsymbol{\pi}}$ built on $2^{\sum_{i=1}^{M} b_i}$ samples, i.e.,*

$$\mathbb{E}_{\boldsymbol{\Delta}^{[1]} \sim q_{\boldsymbol{\pi}}^{[1]}} \left[ \ldots \mathbb{E}_{\boldsymbol{\Delta}^{[M]} \sim q_{\boldsymbol{\pi}}^{[M]}} \left[ \zeta(\boldsymbol{\Delta}^{[1]}, \ldots, \boldsymbol{\Delta}^{[M]}) \right] \ldots \right] = \mathbb{E}_{\boldsymbol{\Delta}^{[1:M]} \sim q_{\boldsymbol{\pi}}} \left[ \zeta(\boldsymbol{\Delta}^{[1:M]}) \right].$$

*Proof.* To show that the above is true it is sufficient to write down these expectations:

$$\mathbb{E}_{\boldsymbol{\Delta}^{[1]} \sim q_{\boldsymbol{\pi}}^{[1]}} \left[ \ldots \mathbb{E}_{\boldsymbol{\Delta}^{[M]} \sim q_{\boldsymbol{\pi}}^{[M]}} \left[ \zeta(\boldsymbol{\Delta}^{[1]}, \ldots, \boldsymbol{\Delta}^{[M]}) \right] \ldots \right] \tag{44}$$

$$= \sum_{k_1=1}^{2^{b_1}} \frac{q_{\phi}(\boldsymbol{\Delta}_{k_1}^{[1]})/p_{\theta}(\boldsymbol{\Delta}_{k_1}^{[1]})}{\sum_{l_1} q_{\phi}(\boldsymbol{\Delta}_{l_1}^{[1]})/p_{\theta}(\boldsymbol{\Delta}_{l_1}^{[1]})} \sum_{k_2=1}^{2^{b_2}} \ldots \sum_{k_M=1}^{2^{b_M}} \frac{q_{\phi}(\boldsymbol{\Delta}_{k_M}^{[M]})/p_{\theta}(\boldsymbol{\Delta}_{k_M}^{[M]})}{\sum_{l_M} q_{\phi}(\boldsymbol{\Delta}_{l_M}^{[M]})/p_{\theta}(\boldsymbol{\Delta}_{l_M}^{[M]})} \zeta(\boldsymbol{\Delta}_{k_1}^{[1]}, \ldots, \boldsymbol{\Delta}_{k_M}^{[M]}) \tag{45}$$

$$= \sum_{k_1=1}^{2^{b_1}} \sum_{k_2=1}^{2^{b_2}} \ldots \sum_{k_M=1}^{2^{b_M}} \frac{q_{\phi}(\boldsymbol{\Delta}_{k_1}^{[1]})/p_{\theta}(\boldsymbol{\Delta}_{k_1}^{[1]})}{\sum_{l_1} q_{\phi}(\boldsymbol{\Delta}_{l_1}^{[1]})/p_{\theta}(\boldsymbol{\Delta}_{l_1}^{[1]})} \ldots \frac{q_{\phi}(\boldsymbol{\Delta}_{k_M}^{[M]})/p_{\theta}(\boldsymbol{\Delta}_{k_M}^{[M]})}{\sum_{l_M} q_{\phi}(\boldsymbol{\Delta}_{l_M}^{[M]})/p_{\theta}(\boldsymbol{\Delta}_{l_M}^{[M]})} \zeta(\boldsymbol{\Delta}_{k_1}^{[1]}, \ldots, \boldsymbol{\Delta}_{k_M}^{[M]}) \tag{46}$$

$$= \sum_{k_1=1}^{2^{b_1}} \sum_{k_2=1}^{2^{b_2}} \ldots \sum_{k_M=1}^{2^{b_M}} \frac{\frac{q_{\phi}(\boldsymbol{\Delta}_{k_1}^{[1]}) \ldots q_{\phi}(\boldsymbol{\Delta}_{k_M}^{[M]})}{p_{\theta}(\boldsymbol{\Delta}_{k_1}^{[1]}) \ldots p_{\theta}(\boldsymbol{\Delta}_{k_M}^{[M]})}}{\sum_{l_1} \ldots \sum_{l_M} \frac{q_{\phi}(\boldsymbol{\Delta}_{l_1}^{[1]}) \ldots q_{\phi}(\boldsymbol{\Delta}_{l_M}^{[M]})}{p_{\theta}(\boldsymbol{\Delta}_{l_1}^{[1]}) \ldots p_{\theta}(\boldsymbol{\Delta}_{l_M}^{[M]})}} \zeta(\boldsymbol{\Delta}_{k_1}^{[1]}, \ldots, \boldsymbol{\Delta}_{k_M}^{[M]}) \tag{47}$$

$$= \sum_{k_*=1}^{2^{b_1+\ldots+b_M}} \frac{\frac{q_{\phi}(\boldsymbol{\Delta}_{k_*}^{[1:M]})}{p_{\theta}(\boldsymbol{\Delta}_{k_*}^{[1:M]})}}{\sum_{l_*=1}^{2^{b_1+\ldots+b_M}} \frac{q_{\phi}(\boldsymbol{\Delta}_{l_*}^{[1:M]})}{p_{\theta}(\boldsymbol{\Delta}_{l_*}^{[1:M]})}} \zeta(\boldsymbol{\Delta}_{k_*}^{[1:M]}) \tag{48}$$

$$= \mathbb{E}_{\boldsymbol{\Delta}^{[1:M]} \sim q_{\boldsymbol{\pi}}} \left[ \zeta(\boldsymbol{\Delta}^{[1:M]}) \right], \tag{49}$$

where $k_* = (k_1, k_2, \ldots, k_M)$ and $l_* = (l_1, l_2, \ldots, l_M)$ are multi-indexes. The line (48) is because distributions $q_{\phi}^{[1]}, \ldots, q_{\phi}^{[M]}$ and $p_{\theta}^{[1]}, \ldots, p_{\theta}^{[M]}$ for different parts of the model are parameterized by the model updates / learnt parameters, and hence, are independent given these updates / parameters. $\square$

# B ADDITIONAL DISCUSSION AND ALGORITHMS

## B.1 ADDITIONAL DISCUSSION

**Computational overhead** `DP-REC` does introduce additional computational requirements compared to standard `FedAvg` and compared to `DP-FedAvg`. Every line in Algorithm 3 involves some additional computation due to the `REC` compression. More specifically, `REC` involves locally sampling $K$ i.i.d. normal samples of dimensionality of the update. After clipping updates, `REC` computes the importance samples $\alpha_k$, essentially calculating log-probabilities of the samples under two Gaussian distributions. Finally, sampling from the categorical distribution $\tilde{q}_{\boldsymbol{\pi}}$, which is relatively cheap since we perform per-tensor compression. It is difficult to benchmark the real-world impact of this additional computational overhead given heterogeneous hardware. A practical implementation would sample the standard-normal samples during training (or even during idle-time), as well as the Gumbel-samples for the categorical distribution. The log-probabilities of the prior can equally be pre-computed. The index-selection procedure can be parallelized if the hardware supports it. In our simulated

environment on a RTX2080 GPU, which runs everything sequentially (training, then per-tensor Gaussian sampling, per-tensor $\alpha$ computation and index-selection), for the Cifar10 experiment with $K = 2^7$, a single-client's epoch has a roughly 70%:30% ratio of training-to-compression, with some variance due to the different local data-set sizes.

**Increasing accuracy at the cost of communication** There is a noticeable accuracy gap between `DP-REC` and `DP-FedAvg`. A reasonable question to ask is whether it is possible to trade higher communication spending for better accuracy? Unfortunately, it is not as simple. The reason for observing the gap lies not in compression but rather in privacy accounting. Figure 2 shows empirically that increasing bit-width has marginal returns, up until doing no compression at all. Any configuration in terms of higher bit-widths or more fine-grained vector quantization can be expected to perform between 7-bit per-tensor quantization and no-compression. Compression-only experiments in Appendix D.4 also corroborate this point, as the non-private compressed model achieves performance much closer to FedAvg. Expressing the divergence bound of discrete distributions through continuous distributions leads to nearly double the amount of noise necessary for an equivalent guarantee compared to the normal continuous Gaussian mechanism. Thus, we would need to relax privacy to close the accuracy gap. This bound with overhead appears to be tight too, judging by some of our synthetic experiments measuring how close it comes to the divergence computed from actual discrete distributions. However, there is still a possible way to reduce the accuracy difference by employing stronger privacy amplification (e.g. adding a secure shuffler), which will counter the overhead of the bound. This effect is seen in StackOverflow dataset, where subsampling amplification is stronger due to a large number of users and the accuracy gap between `DP-FedAvg` and `DP-REC` is drastically smaller.

## B.2 ADDITIONAL ALGORITHMS

### B.2.1 SERVER-TO-CLIENT MESSAGE COMPRESSION

Algorithm 5 and Algorithm 6 formalize the process described in Section 2.5.

---

**Algorithm 5** The server side algorithm for the compression of server-to-client communication. $R_s^{(t)}$ is the client-chosen shared random seed for round $t$. *dec.* describes Alg. 4. $O^{(t)}$ describes the sever-side optimizer including its state (*e.g.* momenta)

---

$H_s = \{(R', O^{(0)})\}\, \forall s \in S$  ▷ History with model initial seed $R'$
**for** $t \in \{0 \ldots T\}$ **do**
   Sample set $S'$ of participating clients
   $\mathcal{M} \leftarrow \{\}$         ▷ Round memory
   **for** $s \in S'$ in parallel **do**
      $msg_s \leftarrow create\_message\_for\_client(s, H_s)$
      *Send* msg$_s$ to client $s$
      *Receive* $k_s^*, R_s^{(t)}$ ▷ client-chosen index, seed
      $\mathcal{M} \leftarrow \mathcal{M} \cup \{(k_s^*, R_s^{(t)})\}$
   **end for**
   **for** $s \in S$ **do**
      $H_s \leftarrow update\_history(s, S', H_s, \mathcal{M}))$
   **end for**
   $\Delta^{(t)} \leftarrow \frac{1}{S'} \sum_{k^* \in \mathcal{M}} dec.(R^{(t)}, k^*)$
   $\mathbf{w}^{(t+1)} \leftarrow \mathbf{w}^{(t)} - O^{(t)}(\Delta^{(t)})$
**end for**

**procedure** CREATE MESSAGE FOR CLIENT$(s, H_s)$
   **if** $size(H_s) > size(\mathbf{w}^{(t)}) + size(O^{(t)})$ **then**
      $msg_s \leftarrow (\mathbf{w}^{(t)}, O^{(t)})$
   **else**
      $msg_s \leftarrow H_s$
   **end if**
   $H_s \leftarrow \{\}$       ▷ Reset client history
   **return** $msg_s$
**end procedure**

**procedure** UPDATE HISTORY$(s, S', H_s, \mathcal{M})$
   **if** $s \in S'$ **then**
      $H_s \leftarrow H_s \cup \{\mathcal{M}\backslash\{(k_s^*, R_s^{(t)})\})\}$
   **else**
      $H_s \leftarrow H_s \cup \{\mathcal{M}\}$
   **end if**
   **return** $H_s$
**end procedure**

---

**Algorithm 6** The client side algorithm for the decompression of server-to-client communication. *dec.* describes Alg. 4

---

**if** $msg_s = (\mathbf{w}^{(t)}, O)$ **then**
   $\mathbf{w}^{(t)} \leftarrow \mathbf{w}^{(t)}$
   $O_s^{(t)} \leftarrow O^{(t)}$
**else**
   $H \leftarrow msg_s$
   $\mathbf{w}^{(t)} \leftarrow \mathbf{w}_s^{(prev)}$ ▷ Last-known server model
   **for** $\mathcal{M} \in H$ **do**
      **if** $\mathcal{M} = (R', O^0)$ **then**
         $\mathbf{w}^{(t)} \leftarrow initialize(R')$
         $O_s \leftarrow O^{(0)}$
      **else**
         $\Delta \leftarrow \frac{1}{|\mathcal{M}|} \sum_{(k,R) \in \mathcal{M}} dec.(R, k)$
         $\mathbf{w}^{(t)} \leftarrow \mathbf{w}^{(t)} - O_s(\Delta)$
      **end if**
   **end for**
**end if**
**return** $\mathbf{w}^{(t)}$

---

### B.2.2 ALGORITHM FOR ACCOUNTING PRIVACY IN DP-REC

Algorithm 7 describes how we compute $\varepsilon$ for a target $\delta$ in DP-REC in general. More specifically, in our experiments, $p_{\boldsymbol{\theta}}^{(t)} = \mathcal{N}(\mathbf{0}, \sigma^2 \mathbf{I})$ and $q_{\boldsymbol{\phi}}^{(t)} = \mathcal{N}(\phi_q, \sigma^2 \mathbf{I})$. For such distributions, the Rényi divergence between them evaluates to

$$\mathcal{D}_\lambda\left(q_{\boldsymbol{\phi}}^{(t)} \| p_{\boldsymbol{\theta}}^{(t)}\right) = \frac{\lambda}{2\sigma^2} \|\phi_q\|_2^2. \tag{50}$$

Thus, for a given $\lambda$ and $\sigma$, bounding this divergence corresponds to clipping the norm of $\phi_q$, *i.e.*, clipping $\|\phi_q\|_2$ to $C$ corresponds to $\mathcal{D}_\lambda\left(q_\phi^{(t)}\|p_\theta^{(t)}\right) \leq \frac{\lambda C^2}{2\sigma^2}$, $\forall\phi,\theta$. In order to allow for privacy amplification with subsampling, we also need to bound the Rényi divergence between the prior $p_\theta^{(t)}$ and the mixture $\frac{B}{N}q_\phi^t + \frac{N-B}{N}p_\theta^{(t)}$ where $N$ corresponds to the number of participants in the federation and $B/N$ to the subsampling rate (Abadi et al., 2016; Mironov et al., 2019). In the general case, we have to consider the maximum over the two possible directions:

$$\mathcal{D}_\lambda\left(\frac{N-B}{N}p_\theta^{(t)} + \frac{B}{N}q_\phi^{(t)}\middle\| p_\theta^{(t)}\right), \qquad \mathcal{D}_\lambda\left(p_\theta^{(t)}\middle\| \frac{N-B}{N}p_\theta^{(t)} + \frac{B}{N}q_\phi^{(t)}\right). \tag{51}$$

However, the calculation can be simplified, at least for certain choices of the distributions $p_\theta^{(t)}, q_\phi^{(t)}$, as (Mironov et al., 2019) show that

$$\mathcal{D}_\lambda\left(\frac{N-B}{N}p_\theta^{(t)} + \frac{B}{N}q_\phi^{(t)}\middle\| p_\theta^{(t)}\right) \geq \mathcal{D}_\lambda\left(p_\theta^{(t)}\middle\| \frac{N-B}{N}p_\theta^{(t)} + \frac{B}{N}q_\phi^{(t)}\right), \tag{52}$$

allowing us to focus on the first term. Furthermore, again following (Abadi et al., 2016; Mironov et al., 2019), this divergence can be simplified for a general mixture with weights $\alpha$ and $(1-\alpha)$, our specific choice of $q_\phi^{(t)}$ and $p_\theta^{(t)}$, and for integer $\lambda$:

$$\mathcal{D}_\lambda\left((1-\alpha)p_\theta^{(t)} + \alpha q_\phi^{(t)}\middle\| p_\theta^{(t)}\right) = \frac{1}{\lambda-1}\log\left(\mathbb{E}_{p_\theta^{(t)}}\left[\left(\frac{(1-\alpha)p_\theta^{(t)} + \alpha q_\phi^{(t)}}{p_\theta^{(t)}}\right)^\lambda\right]\right) \tag{53}$$

$$= \frac{1}{\lambda-1}\log\left(\mathbb{E}_{k\sim\mathcal{B}(\lambda,\alpha)}\left[\mathbb{E}_{p_\theta^{(t)}}\left[\left(\frac{q_\phi^{(t)}}{p_\theta^{(t)}}\right)^k\right]\right]\right) \tag{54}$$

$$\leq \frac{1}{\lambda-1}\log\left(\mathbb{E}_{k\sim\mathcal{B}(\lambda,B/N)}\left[e^{\frac{k^2-k}{2\sigma^2}C^2}\right]\right). \tag{55}$$

This allows us to readily calculate all of the terms in Algorithm 7.

---

**Algorithm 7** Privacy accounting for `DP-REC` with subsampling for privacy amplification. $\Lambda$ are the Rényi orders $\lambda > 1$ that will be considered and $b$ are the total number of bits used to represent the neural network update. Furthermore, $T$ are the total communication rounds for training, $B$ is the number of users considered on each round and $N$ is the total number of users in the federation. $\delta$ is the target probability of DP failing to provide a guarantee.

---

$\rho^{(0)} \leftarrow 0$
$\hat{k}_\lambda^{(0)} \leftarrow 0, \ \forall\lambda\in\Lambda$
$\hat{m}_\lambda^{(0)} \leftarrow 0, \ \forall\lambda\in\Lambda$
**for** $t \leftarrow 1,\ldots,TB$ **do**
$\quad \rho^{(t)} \leftarrow \rho^{(t-1)} + \max_{\theta,\phi} e^{\mathcal{D}_2\left(q_\phi^{(t)}\|p_\theta^{(t)}\right)}$
$\quad$ **for** $\lambda\in\Lambda$ **do**
$\qquad \hat{k}_\lambda^{(t)} \leftarrow \hat{k}_\lambda^{(t-1)} + \max\begin{cases}\max_{\theta,\phi}\mathcal{D}_\lambda\left(\frac{N-1}{N}p_\theta^{(t)} + \frac{1}{N}q_\phi^{(t)}\middle\| p_\theta^{(t)}\right) \\ \max_{\theta,\phi}\mathcal{D}_\lambda\left(p_\theta^{(t)}\middle\| \frac{N-1}{N}p_\theta^{(t)} + \frac{1}{N}q_\phi^{(t)}\right)\end{cases}$
$\qquad \hat{m}_\lambda^{(t)} \leftarrow \hat{m}_\lambda^{(t-1)} + \max\begin{cases}\max_{\theta,\phi}\mathcal{D}_{\lambda+1}\left(\frac{N-1}{N}p_\theta^{(t)} + \frac{1}{N}q_\phi^{(t)}\middle\| p_\theta^{(t)}\right) \\ \max_{\theta,\phi}\mathcal{D}_{\lambda+1}\left(p_\theta^{(t)}\middle\| \frac{N-1}{N}p_\theta^{(t)} + \frac{1}{N}q_\phi^{(t)}\right)\end{cases}$
$\quad$ **end for**
**end for**
$\varepsilon \leftarrow \min_\lambda\left(\frac{\lambda-1}{\lambda}\hat{k}_\lambda^{(TB)} + \hat{m}_\lambda^{(TB)} - \frac{1}{\lambda}\log\left(\delta - \frac{12}{2^b}\rho^{(TB)}\right)\right)$

---

### B.2.3 OVERALL ALGORITHM FOR FEDERATED TRAINING WITH DP-REC

---

**Algorithm 8** Overall federated training pipeline when using DP-REC. $S$ corresponds to all clients in the federation and $s$ to a specific client. $C$ corresponds to the desired sensitivity.

---

**procedure** SERVER TRAINING$(T, \sigma)$
$\quad H_s = \{(R', O^{(0)})\} \, \forall s \in S$ $\qquad\qquad\qquad\qquad$ ▷ History with model initial seed $R'$
$\quad$**for** $t \in \{0 \dots T\}$ **do**
$\quad\quad S' \leftarrow$ Sample set of participating clients with replacement
$\quad\quad \mathcal{M} \leftarrow \{\}$ $\qquad\qquad\qquad\qquad\qquad\qquad\qquad\qquad\qquad$ ▷ Round memory
$\quad\quad$**for** $s \in S'$ in parallel **do**
$\quad\quad\quad msg_s \leftarrow create\_message\_for\_client(s, H_s)$
$\quad\quad\quad k_s^*, R_s^{(t)} \leftarrow client\_training(msg_s)$
$\quad\quad\quad \mathcal{M} \leftarrow \mathcal{M} \cup \{(k_s^*, R_s^{(t)})\}$
$\quad\quad$**end for**
$\quad\quad$**for** $s \in S$ **do**
$\quad\quad\quad H_s \leftarrow update\_history(s, S', H_s, \mathcal{M}))$
$\quad\quad$**end for**
$\quad\quad \Delta^{(t)} \leftarrow \frac{1}{S'} \sum_{k^* \in \mathcal{M}} dec.(R^{(t)}, k^*)$
$\quad\quad \mathbf{w}^{(t+1)} \leftarrow \mathbf{w}^{(t)} - O^{(t)}(\Delta^{(t)})$
$\quad$**end for**
$\quad \varepsilon \leftarrow$ compute $\varepsilon$ guarantee for a given $\delta$. $\qquad\qquad\qquad\qquad$ ▷ Using Alg. 7
**end procedure**

**procedure** CLIENT TRAINING$(msg_s)$
$\quad \mathbf{w}^{(t)} \leftarrow receive\_message(msg_s)$ $\qquad\qquad\qquad\qquad\qquad$ ▷ Refers to Alg. 6
$\quad \mathbf{w}_s^{(t)} \leftarrow \mathbf{w}^{(t)}$
$\quad$**for** local epoch $e \in \{1, \dots, E\}$ **do**
$\quad\quad$**for** batch $b \in B$ **do**
$\quad\quad\quad \mathbf{w}_s^{(t)} \leftarrow \mathbf{w}_s^{(t)} - \nabla\ell(\mathbf{w}_s^{(t)}, b)$ $\qquad\qquad\qquad$ ▷ $\ell$ corresponds to the local loss
$\quad\quad$**end for**
$\quad$**end for**
$\quad \phi_s^{(t)} \leftarrow \mathbf{w}_s^{(t)} - \mathbf{w}^{(t)}$
$\quad \hat{\phi}_s^{(t)} = \phi_s^{(t)} \min(1, C/\|\phi_s^{(t)}\|_2)$
$\quad R_s^{(t)} \leftarrow$ Choose a random seed
$\quad k_s^* \leftarrow enc.(\hat{\phi}_s^{(t)}, R_s^{(t)})$ $\qquad\qquad\qquad\qquad\qquad\qquad\qquad$ ▷ Using Alg. 1
$\quad$**return** $k_s^*, R_s^{(t)}$
**end procedure**

---

## C EXPERIMENTAL DETAILS

The experiments in the main text were performed on two Nvidia RTX 2080Ti GPUs, as well on several Nvidia Tesla V100 GPU's available on an internal cluster over the span of two weeks.

### C.1 DATASETS & MODELS

**MNIST** For MNIST, we consider a LeNet-5 (LeCun et al., 1998) model trained on a federated version of the original 50k training images. We split the data across 100 clients in a non-i.i.d. way where the label proportions on each client are determined by sampling a Dirichlet distribution with concentration $\alpha = 1.0$ (Hsu et al., 2019). For evaluation, we assume the standard validation split of MNIST to be available at the server. We run all experiments four times with different seeds, except the one with DP-REC and $\varepsilon = 3$ which we run with eleven seeds, as it had an "outlier" run which skewed the average and increased considerably the standard error.

**FEMNIST** FEMNIST is a federated version of the extended MNIST (EMNIST) dataset (Cohen et al., 2017). It consists of MNIST-like images of handwritten letters and digits belonging to one

Table 2: Federated training sets

| Dataset | Number of devices | Total samples | Samples per device | |
|---------|-------------------|---------------|------|------|
| | | | mean | std |
| MNIST | 100 | $50,000$ | 500.00 | 73.10 |
| FEMNIST | 3500 | $705,595$ | 201.60 | 78.92 |
| Shakespeare | 660 | $3,678,451$ | 5573.41 | 6460.77 |
| StackOverflow | $342,477$ | $135,818,730$ | 396.58 | 1278.94 |

of 62 classes. The federated nature of the dataset is naturally determined by the writer for a given datapoint. Additionally, the size of the individual clients' datasets differ significantly. In the literature, there are two versions of this dataset used for experimentation. Originally published by (Caldas et al., 2018), their published code[1] provides a recipe to pre-process the dataset into the federated version. Unfortunately, however, the statistics reported in the paper do not align with the result of this recipe. We repeat the statistics as we use them for our experiments in Table 2. As mentioned in the main text, some works such as `DDGauss` (Kairouz et al., 2021) use the FEMNIST version provided by tensorflow federated[2]. It consists of a smaller subset of 3400 clients. For the model architecture, we consider the convolutional network described in (Kairouz et al., 2021), albeit without dropout regularization. We run four seeds for `DP-FedAvg` and `DP-REC` with $\varepsilon = 1, 3, 6$.

**Shakespeare** For the Shakespeare dataset we closely follow (Caldas et al., 2018). Each client corresponds to a unique character across the collection of Shakespeare's plays with a minimum number of spoken lines. The non-i.i.d. characteristics of this dataset are due to the different "speaking" styles of the resulting 660 roles. We use the same 2-layer LSTM model as in (Caldas et al., 2018) for this next-character-prediction task (considering a library of 80 characters). Each client predicts the next character following the LSTM encoding of the previous 80 characters. For Table 2 we consider each pair of 80 character plus next character as a single sample. The statistics we report differ markedly from the statistics in (Caldas et al., 2018), as reported on a corresponding issue raised in their code base[3]. We run four seeds for both `DP-FedAvg` and `DP-REC`.

**StackOverflow** The StackOverflow dataset (TFF Authors, 2019) consists of a collection of questions and answers posted on the StackOverflow website during a certain time window. Each user of that website who posted there in that time-frame is considered a client with their aggregated posts as the client's dataset. Here, we consider the task of tag prediction described in (Reddi et al., 2020). Associated with each posting (irrespective of whether it is a question or an answer) is associated at least one of 500 tags. Each client is therefore performing one-vs-all classification corresponding to 500 binary classifications. We pre-process each post by creating a bag-of-words representation of the $10,000$ most frequent words, normalized to 1. As model, we consider logistic regression. Learning curves in the main paper were created by selecting the first $10,000$ data-points when iterating over a shuffled list of hold out clients. As noted in the main text, each run selected a different seed for shuffling, resulting in non directly comparable learning curves. For the results in Table 1, the final model was evaluated on all $16,586,035$ datapoints across all hold out client. Note that with $1,500$ communication rounds and 60 clients per round, only $\sim 26\%$ of all clients participate in training. We run two seeds for both `DP-FedAvg` and `DP-REC`.

### C.2 Hyperparameters

For all of the experiments in the main text we used 7-bit per tensor quantization for `DP-REC`. This was determined after the ablation study on the FEMNIST dataset, shown at Figure 2. The only difference was at the StackOverflow experiment where, to have a reasonable $\varepsilon$ DP guarantee, we used 5 quantization groups for the weight matrix of the logistic regression model (each with $10^6$ entries) and did per-tensor quantization for the biases (*i.e.*, we had 6 quantization groups in total). As we

---

[1]`https://github.com/TalwalkarLab/leaf`
[2]`https://www.tensorflow.org/federated/api_docs/python/tff/simulation/datasets/emnist`
[3]`https://github.com/TalwalkarLab/leaf/issues/13`

mentioned in the main text, for DP-REC we performed sampling with replacement to select clients for each round on all tasks, since the improved privacy amplification was beneficial. For DP-FedAvg we use the traditional sampling without replacement on each round but with replacement across rounds.

**Tuning privacy hyperparameters** Privacy guarantees depend essentially on the ratio of the clipping threshold and the prior noise scale $\sigma$. For all experiments, we fixed the ratio, determined by our accounting upfront, in order to ensure a chosen $\varepsilon$ at the end of the experiment. More specifically, for DP-FedAvg we tune the clipping threshold for each task and pick the appropriate noise scale for a given $\varepsilon$ guarantee. For DP-REC the clipping threshold was a multiplicative factor of the standard deviation of the prior over the deltas, i.e., $c\sigma$, and $c$ was tuned in order to yield a specific $\varepsilon$ guarantee. The free parameter that we thus optimize is $\sigma$. For $\sigma$, we considered values ranging from $10^{-5}$ up to 1, finding the right order of magnitude and then fine-tuning within that order if necessary (e.g. considering 0.001, 0.003, 0.005, etc.), based on validation performance. Of course, in a practical deployment such tuning would need to be taken into account when computing final privacy parameters, as discussed by Abadi et al. (2016, Appendix D).

**MNIST** For this task we used SGD with a learning rate of 0.01 for the client optimizer and Adam (Kingma & Ba, 2014) with a learning rate of $2e - 3$ for the server optimizer for all of the experiments. The $\beta_1, \beta_2$ parameters of Adam were kept at the default values (0.9 and 0.999) and we trained for 1k global communication rounds rounds. We used 10 clients on each round, where each client performed 1 local epoch with a batch size of 20. For DP-FedAvg the clipping threshold was 0.01 whereas for DP-REC the prior standard deviation was fixed to $\sigma = 0.005$. In order to get $\varepsilon = 3, 6$ for DP-FedAvg we used a noise scale of 3.8 and 2.15, whereas for DP-REC we used a $c = 0.5$ and $c = 0.7625$ respectively.

**FEMNIST** For optimization we used SGD with a learning rate of 0.05 locally and SGD with a learning rate of 1.0 globally, i.e., we averaged the local parameters. For all of the methods we sampled 100 clients for each round and each client performed 1 local epoch with a batch size of 20. For DP-FedAvg we trained for 1.5k rounds with a clipping threshold of 0.1 and for DP-REC we found it beneficial to train for 4k rounds (and thus we had to clip more aggressively on each round) and the $\sigma$ was fixed to 0.03. In order to get $\varepsilon$ of 1, 3 and 6 we used a noise scale of 5, 1.85 and 1.15 for DP-FedAvg and a $c$ of 0.75, 1.35 and 1.66 for DP-REC.

**Shakespeare** Both the global and local optimizers for this task were SGD with a learning rate of 1.0. We trained for 200 rounds where on each round we sampled 66 clients and each client performed 1 local epoch with a batch size of 10. For DP-FedAvg the clipping threshold was 0.3 whereas for DP-REC the prior standard deviation was fixed to $\sigma = 0.1$. In order to get a $\varepsilon$ of 3 we used a noise scale of 2.15 for DP-FedAvg and a $c$ of 1.36 for DP-REC.

**StackOverflow** For this task we mainly used the hyperparameters provided at (Andrew et al., 2019). More specifically, for the local optimizer we used SGD with a learning rate of $10^{2.5}$ whereas for the global optimizer we used SGD with a learning rate of $10^{-0.25}$ and a momentum of 0.9. We sampled 60 clients per round and each client performed 1 local epoch with a batch size of 100. We trained the logistic regression model for 1.5k rounds. For DP-FedAvg the clipping threshold was 0.05 whereas for DP-REC the prior standard deviation was $\sigma = 0.05$. For a $\varepsilon$ of 1 we used a noise scale of 0.957 for DP-FedAvg and a $c$ of 1.227 for DP-REC.

## D  ADDITIONAL RESULTS

### D.1  ACCURACY FOR A FIXED PRIVACY BUDGET

Figure 3 shows the accuracy achieved as a function of the privacy budget $\varepsilon$. For a discussion of these results refer to the main text.

### D.2  PRIVATE MEAN ESTIMATION

We ran an experiment to compare DP-REC with the method of Chen et al. (2020) and see the effects of Kashin's representation and shared randomness. Removing privacy from the equation and comparing compression methods head-to-head with 1-bit communication (+ bits for shared randomness, as we

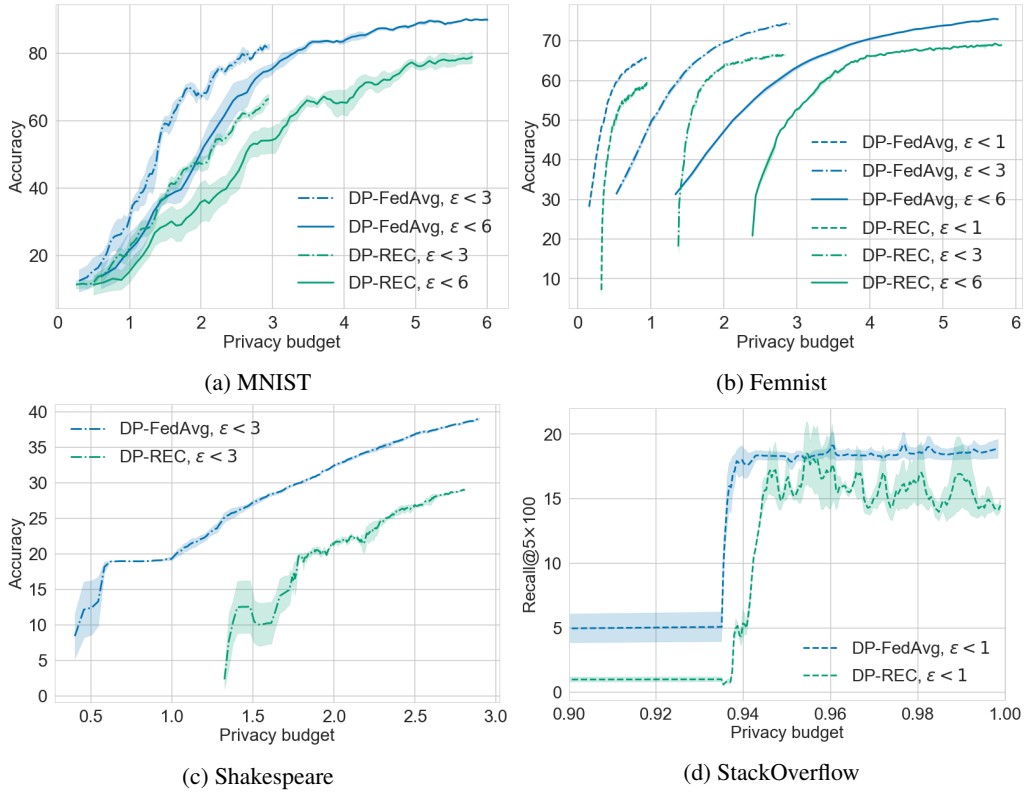

(a) MNIST

(b) Femnist

(c) Shakespeare

(d) StackOverflow

Figure 3: Test accuracy (%) as a function of the privacy budget $\varepsilon$ (for a fixed $\delta = 1/N^{1.1}$).

explain below), we find that (Chen et al., 2020) performs slightly better than `DP-REC` in terms of mean squared error (MSE), when the latter uses poorly informed prior (e.g. a zero-mean Gaussian). When `DP-REC` is equipped with a better prior (e.g. Gaussian with the mean $1/\sqrt{d}$ in all dimensions), it is comparable or outperforms (Chen et al., 2020). For example, estimating a 200-dimensional mean from 10k samples, MSE of (Chen et al., 2020) is 0.07, for `DP-REC` with a poor prior it is $\sim 0.3$, and for `DP-REC` with a better prior it is $\sim 0.03$. There are a few points to elaborate on:

- Shared randomness. Similarly to `DP-REC`, (Chen et al., 2020) have a choice between public randomness (defined by the server) or a shared randomness (defined by clients). As we explained in our paper, the latter is a preferred choice in terms of privacy, but adds a few bits to client-server communication (for the random seed). We found that it also improves performance and that the method of Chen et al. (2020) did not work well in few-bit settings with public randomness (MSE > 10.0 in the above example).

- Prior. `DP-REC` is a method that's more dependent on a good prior. With a poor choice, in one-shot settings like mean estimation, performance can be compromised. However, it makes it more suitable for FL scenarios where prior is gradually improved with every round.

- Adding privacy to the mix. It is important to note that `DP-REC` communication gets somewhat penalized due to the overhead term in our Theorem 1, which for the above setting requires at least 21 bits per message to achieve $\delta < 10^{-5}$. Nonetheless, this is not a problem in FL, since practical communication bit-width would be larger in any case. Moreover, `DP-REC` privacy can be further amplified by implementing the secure shuffler, similarly to Girgis et al. (2020), resulting in even tighter privacy guarantees.

### D.3 ADDITIONAL BASELINES

A curious reader may wonder how would a simple baseline of compressed gradients combined with `DP-FedAvg` fare against our method. Unfortunately, combining `DP-FedAvg` (or differential

privacy in general) with compression is not a straightforward task, which is why it motivates our paper. There are two options: (i) ensure DP, then compress the update; (ii) compress, then ensure DP. Each option has its challenges.

First, consider (i). Adding noise at the client and then compressing the update before transmission, might not allow to calibrate noise to the aggregate and the use tighter composition theorems (e.g., Rényi accountant), degrading the privacy guarantee. This is what essentially led to the development of hybrid methods such as `cpSGD` and `DDGauss`. As this direction was researched in the line of work leading up to `DDGauss`, we take their method as the best representation of such approach. Let us now consider (ii). Once updates are compressed by the client, we cannot add noise directly, because it would negate any compression. Therefore, the client needs to transmit the update first, using secure aggregation to protect against honest-but-curious server. We can compress updates using scalar quantization, but secure aggregation might add communication overhead countering some of the effects of compression (see Bonawitz et al. (2017)). Even without the additional communication overhead of secure aggregation, we empirically observed that such a method can have both worse accuracy and worse communication efficiency than `DP-REC` and represents a rather weak baseline. We ran an experiment on our MNIST task and combined 8-bit scalar quantization of the client updates (in a way that satisfies the desired sensitivity) with `DP-FedAvg` with $\varepsilon = 3$; there we observed that the global accuracy reached $\sim 58\%$ after 1k rounds, which is both smaller than the `DP-REC` result and significantly more expensive in terms of communication. Lastly, if we were to consider vector quantization to get ahead in terms of compression ratios, it would hinder application of secure aggregation, leaving us without any protection against honest-but-curious server (since adding noise is not possible either, as it would nullify compression).

Of course, there exists a number of diverse gradient compression methods that could be considered, and that could be more easily combined with either `SecAgg` or DP, but we leave exploration of these methods for future work.

### D.4 COMPRESSION-ONLY PERFORMANCE OF OUR METHOD

In order to investigate the behaviour of the compression part of `DP-REC`, we performed some experiments on our 4k round FEMNIST task where we omit the clipping part of `DP-REC`. We consider three runs:

1. A baseline of vanilla `FedAvg` on this task without compression,

2. `REC` compression with the same hyperparameters as for the with-DP experiments but without clipping,

3. `REC` compression but with quantization of groups of size 64 (instead of per-tensor quantization),

The results can be found at Table 3, where we also include the results from `DP-REC` for reference. We can see that the compression part of `DP-REC` performs quite well, reaching performance similar to `FedAvg` while significantly reducing the total communication costs. It is worthwhile to note that there is a multitude of other hyperparameters that could be tuned to further refine the compression-only performance, such as: tuning the prior $\sigma$; varying the vector-size over which $\sigma$ is computed; the bit-width $b$; additional application of entropy-coding for the larger number of indices; adaptive $K$ per round. We leave a more detailed compression-only evaluation of our method to future work.

| Method | Accuracy | GB |
|---|---|---|
| DP-REC ($\varepsilon\!=\!1$) | $59.3 \pm 0.1$ | 14.2 |
| DP-REC ($\varepsilon\!=\!3$) | $67.0 \pm 0.1$ | 14.2 |
| DP-REC ($\varepsilon\!=\!6$) | $69.1 \pm 0.1$ | 14.2 |
| FedAvg (no DP / compression) | 86.28 | 690.6 |
| REC (no DP clipping) | 80.69 | 14.2 |
| REC (no DP clipping, $64$) | 84.14 | 332.2 |

Table 3: Compression-only (i.e., no DP) ablation studies, all run for 4k rounds. *was still improving after $4k$ rounds.

