# OpenReview forum: "DP-REC: Private & Communication-Efficient Federated Learning"
_ICLR.cc/2022/Conference — ICLR 2022 Submitted_

### Official Review · Reviewer_qyw3 · 2021-10-31

**Correctness:** 4
**Technical Novelty And Significance:** 3
**Empirical Novelty And Significance:** 2
**Recommendation:** 6
**Confidence:** 4

**Main Review:**

I think the most interesting part of this paper is that it leverages the randomness in the sampling process to achieve differential privacy, and gives privacy bounds. I didn't check the proof in the appendix in detail, but the analysis makes sense to me at the high level.

This work also considers local privacy in addition to client-level privacy. It describes the relations with previous works and the experimental setups reasonably well.

My major concerns are that (1) the recovered model updates at the server side (even without clipping for privacy) are biased wrt the true model updates; (2) the algorithm doesn't support selecting multiple clients at each round, and (3) experiments can be improved.

Regarding (1), I think it is probably fine (and not difficult) to give a convergence bound with a small error term. Regarding (2), if there are many clients in the entire network, picking only one of them at each communication round can make the convergence unstable, or reach a lower accuracy. I am wondering if it is possible to extend the current algorithm by selecting a subset of clients, allowing each of them to run Alg 3 independently (with their own random seeds), and aggregating these model updates at the server (after the server recovers each update with the corresponding random seed)? This will affect both privacy and convergence.

Accuracy drops significantly under the same privacy budget in the experiments (e.g., on the simple MNIST dataset, the accuracy is 10% lower than DP-FedAvg for a large epsilon value), but the compression ratio is very high. DP-REC allows for different compression granularities such as per-parameter clipping. Is it possible that such kind of more fined-grained compression can increase the final accuracy, although transmitting more bits each round?

I think there are multiple potential baselines on compression + DP. E.g., apply DP on top of any gradient compression method. These should be considered or at least discussed in the experiments. The goal of DDGauss is not compression (though it results in quantized model updates). Therefore it may not be a strong compression baseline.
cpSGD, ATOMO (https://arxiv.org/abs/1806.04090) + DP, selecting top-k coordinates (measured by magnitudes)+DP are all reasonable baselines.


Minor:

* I think some notations might be a bit confusing, e.g., w \sim q_{\phi}(w) (the third line in Sec 2.1). It would be more clear to use different notations for random variables and samples. In algorithm 3, can note k \in {1, ..., K}.

* Moving the discussions on the choice of sigma values to the main text could be better.

* Experiments: It would be helpful for readers to further understand the behavior of the proposed method if it also includes results on applying compression alone with a good \sigma without clipping (i.e., not considering privacy budgets). That will demonstrate how good the (biased) compression itself can be.

* What does 'tensor'(generally understood as multi-dimensional arrays) mean in the context of this work? Does it refer to different sets of named parameters in the model?

**Summary Of The Paper:**

This paper proposes a compression scheme for federated learning built upon previous relative entropy encoding works. It then proves that with some small modification (clipping the model updates), the algorithm is inherently differentially private. Empirical evaluation shows that the proposed method can achieve much more communication reduction at the cost of accuracy degradation.

**Summary Of The Review:**

I think the most interesting part of this paper is that it leverages the randomness in the sampling process to achieve differential privacy. My major concerns are that (1) the recovered model updates at the server side (even without clipping for privacy) are biased wrt the true model updates; (2) the algorithm doesn't support selecting multiple clients at each round, and (3) experiments can be improved.


=== update ===

After reading the authors' responses, some of my concerns have been addressed. Therefore, I increase my score to 6.

---

> ### Author Response · Authors · 2021-11-16
> **Response to Reviewer qyw3**
>
> We thank you for the analysis of our work and we are glad that you find our approach interesting! We address your concerns below:
>
> 1. **Bias of the updates.** It is true that REC-compressed model updates are biased. However, as you also suggested, the bias can be bounded (e.g. using Eq. 4). Given the settings of strong privacy (small divergence between the user update distribution and prior), the bias would also be small. And even without privacy constraints, when running our method for compression only, we observe that for an appropriate quantization group size the performance is similar to the standard FedAvg with no compression. Theorem 3 and Eq. 4 could guide the choice of the right number of groups and bit-rates for controlling the bias. Overall, as we haven't observed convergence issues during any of our experiments, we leave the detailed proof for future work.
> 2. **Multiple clients in a round and sub-sampling accounting.** We can happily report that our algorithm does, in fact, select multiple clients in each round. We suspect that our description of privacy accounting amplification led to some misunderstanding. Across all methods in our experiments, we select the same number of clients to compute in parallel (in DP-REC, we sample them with replacement). The reason our amplification discussion refers to single clients is because of the sampling-with-replacement procedure and noise calibration to individual updates rather than aggregates as in DP-FedAvg or DDG.
> 3. **Increasing accuracy at the cost of communication.** Please see our general reply to commonly raised points.
> 4. **Additional baselines.** Unfortunately, combining DP-FedAvg (or DP in general) with compression is not a straightforward task, which is why it motivates our paper. There are two options: (i) ensure DP, then compress the update; (ii) compress, then ensure DP. Each option has its challenges. First, consider (i). Adding noise at the client and then compressing the update before transmission, might not allow to calibrate noise to the aggregate and use tighter composition theorems (e.g., Rényi accountant), degrading the privacy guarantee. This is what essentially led to the development of hybrid methods such as cpSGD and DDGauss. As this direction was researched in the line of work leading up to DDGauss, we take their method as the best representation of such approach (and this is why we don't compare to cpSGD). Let us now consider (ii). Once updates are compressed, clients cannot add noise directly, because it would negate any compression. At the same time, we cannot transmit them in the clear, because we want to protect against honest-but-curious server. So the client has to use secure aggregation (SecAgg). We can compress updates using scalar quantization, but SecAgg might add communication overhead countering some of the effects of compression (see [1]). Even without this communication overhead of SecAgg, we empirically observed that such a method can have both worse accuracy and worse communication cost than DP-REC and represents a rather weak baseline. We ran an experiment on MNIST and combined 8-bit scalar quantization of the client updates (in a way that satisfies the desired sensitivity) with DP-FedAvg with $\varepsilon=3$; we observed that the global accuracy reached $\sim58\\%$ after 1k rounds, which is both smaller than the DP-REC result and much more expensive in terms of communication. If we were to use vector quantization to get ahead in terms of compression ratios, it would hinder application of secure aggregation, leaving us without any protection against honest-but-curious server (since adding noise is not possible either, as it would nullify compression). Finally, we were not familiar with ATOMO at the time of writing the paper, but we suspect this direction could be worth a separate work and we could certainly look into it in the future. For instance, we would need to answer similar questions of combining it with SecAgg/DP, e.g. how to quantize or obfuscate atoms to achieve a better trade-off than scalar quantization.
> 5. **Compression-only experiments.** We chose FEMNIST to provide some compression-only experiments. Please see the new Appendix D.4 for results and discussion. We include FedAvg without compression (reaching $86.28\\%$), compression with the same hyperparameters as in DP experiments ($80.69\\%$); compression with quantization of groups of size 64 ($84.14\\%$). In short, the non-private versions can reach performance close to FedAvg, while still greatly reducing communication. Please note that there are multiple hyperparameters we could tune to further refine performance, but the focus of our paper was explicitly DP with compression, so we leave a more detailed compression-only evaluation to future work.
>
> ---
> [1] Bonawitz, K., et al. Practical secure aggregation for privacy-preserving machine learning. In *Proceedings of the 2017 ACM SIGSAC Conference on Computer and Communications Security*, 2017.

---

> ### Author Response · Authors · 2021-11-16
> **Additional Response to Minor Comments by Reviewer qyw3**
>
> Minor comments:
>
> 1. **Notation.** We will try to clarify our notation in the updated version of the paper after the rebuttal phase (we would like to avoid the notation change during the rebuttal in order not to confuse reviewers and complicate the discussion).
> 2. **Discussion on the choice of $\sigma$.** We did not include that discussion in the main text due to the strict page limit, and unfortunately, we still couldn't find enough space in the revision.
> 3. **Terminology.** In the context of our work, "tensor" also refers to multi-dimensional arrays, and in particular, parameters of a single layer. We speak of tensors rather than matrices or parameters for the purpose of generality, for example, including parameters of higher-dimensional convolution layers.

---

> ### Author Response · Authors · 2021-11-29
> **Response to reviewer qyw3**
>
> Dear reviewer,
>
> we hope to have addressed your concerns appropriately through our rebuttal. As the discussion period is ending soon, we would appreciate if you could acknowledge our rebuttal and let us know if you have any further questions. Thank you very much for your time and reviewing our work.

---

### Official Review · Reviewer_Kc1E · 2021-11-02

**Correctness:** 4
**Technical Novelty And Significance:** 3
**Empirical Novelty And Significance:** 3
**Recommendation:** 8
**Confidence:** 3

**Main Review:**

**Strengths**
1) The Bayesian approach taken by this paper is different from the worst-case guarantees provided in many papers studying a similar problem (Agarwal 18, Chen 20, Girgis 20). The promising results in this paper could therefore a inspire a new line of research in this area.
2) The paper introduces plenty of new ideas to the federated learning community, both in the main algorithm and its analysis.
3) Experiments validate that the proposed algorithm provides very high compression in the high privacy regime ($\varepsilon < 1$).

**Weaknesses**
1) Computational complexity: The algorithm in its general form requires $b$ samples to be drawn from a high-dimensional distribution. This is computationally costlier than many other known methods.

2) Given that Chen 2020 is known to be optimal under joint privacy and communication constraints, a *theoretical* comparison with results in Chen would have to theory of this paper. This could have been done for both *good* and *bad* prior.

**Summary Of The Paper:**

The paper proposes a differentially private and communication-efficient method to aggregate the client updates in federated learning. The method is based on a recently proposed compression technique relative entropy coding. The authors further modify this technique to satisfy differential privacy guarantees and perform various experiments to back their claims.

**Summary Of The Review:**

Overall, I like this paper and recommend it be accepted.

---

> ### Author Response · Authors · 2021-11-16
> **Response to Reviewer Kc1E**
>
> We thank you for the positive review. We are especially proud that you believe our paper has the potential to inspire a new line of research and that you praise the novelty and abundance of ideas for the FL community. Let us provide some short additional comments on the weaknesses you have identified:
>
> 1. **Computational complexity.** You observe correctly that the method increases computational complexity on-device compared to other methods due to drawing a number of high-dimensional samples. In practice, however, this might not be an issue. These samples do not require knowledge of the model or data and can be drawn independent of training, i.e. before training, in-parallel or generally when the device is idle between rounds. In our simulated experiments, the real-world overhead did not appear to be as large as suggested by theoretical complexity, with about $70\\%$ of time spent on training and only $30\\%$ on compression (including the whole REC-compression pipeline) for CIFAR10.
> 2. **Theoretical comparison with (Chen et al., 2020).** Indeed, analyzing theoretical properties of the method, such as bounds on mean estimation, and comparing to (Chen et al., 2020) would be an interesting direction to pursue. For instance, using additional inequalities from (Agapiou et al., 2017) it should be possible to construct a similar, but expected-case bound on the mean squared error for data within a unit ball. As it was not the main focus of the present paper, however, we are planning to leave the detailed investigation for a separate future work.

---

> > ### Comment · Reviewer_Kc1E · 2021-11-29
> > **Response to the Rebuttal**
> >
> > I thank the authors for further clarification. After going through the other reviews and the rebuttal, I stand by my original review.

---

### Official Review · Reviewer_Uy67 · 2021-11-03

**Correctness:** 4
**Technical Novelty And Significance:** 3
**Empirical Novelty And Significance:** 3
**Recommendation:** 8
**Confidence:** 4

**Main Review:**

Pros:

The idea of applying REC to compress a (distributed) DP mechanism is novel and elegant, and the connection between the communication budges and the privacy guarantees are also interesting. The comparisons with previous works and the experiments, in my opinion, are adequate. I also appreciate the authors' discussions on whether DP-REC is compatible with other privacy amplification techniques.

Cons:


1. My major concern is the correctness of "per-parameter compression". All of the theorems, including the privacy guarantee and the communication bound, are derived for the network-wise compression, and the authors claim that one can directly apply these results to parameter-wise compression since both the prior distribution $p_\theta$ and the target distribution $q_\phi$ are product measures. However, I am not very sure about this statement. Though the target distribution $q_\phi$ is a product measure, the law of the outcome of DP-REC (Algorithm 3) $q_{\tilde{\pi}}$ may not be of a product form. Similarly, I don't see how $\pi_k$ in Algorithm 3 can be decomposed into a product form.  Therefore, in general, I am not so sure about whether equation (4) will still hold if we decompose it coordinate-wisely.

Since the main theorem is based on equation (4) and all experiments are carried out with per-tensor compression, I think it is important to give a formal statement and proof to show that the results also hold for per-parameter compression.


2. It would be good if the authors can provide a detailed end-to-end algorithm for the overall FL framework, including how to pick parameters such as $b$ and $\sigma$.


3. When comparing with DDG on the FMNIST dataset, it seems that the accuracy of DP_REC is still significantly lower than the accuracy of DDG. Is it possible to increase the communication budgets and reduce the gap?

**Summary Of The Paper:**

This paper introduces a compression and privatization technique to federated learning based on Relative Entropy Coding (REC). For each round, client $s$ aims to privatize its local model update $w_s$ by releasing $\Delta_s = w_s + N(0, \sigma I)$. However, instead of directly adding Gaussian noise, the client first picks $K$ random vectors from a prior distribution (which is independent of $w_s$) with shared randomness and then performs importance sampling according to the law of $w_s + N(0, \sigma I)$. It can be shown that as long as $K$ large enough (i.e., when $\log_2 K$ is much larger than the Renyi divergence of the prior and the target distributions), the sample obtained from importance sampling has a law close to the target distribution (i.e. the Gaussian perturbation) and thus can potentially preserve privacy. Moreover, the communication needed is $\log_2 K$ bits per round per client. By accounting the privacy loss over $T$ rounds, the authors characterize the overall privacy guarantees, which have a clean form and connect to the communication budgets nicely.

**Summary Of The Review:**

The paper is well-written and the proposed idea is novel. Discussion and comparison with related works are adequate. However, my major concern is the correctness of the per-parameter compression. If the authors can clarify this point, I am happy to further increase my score.

---

> ### Author Response · Authors · 2021-11-16
> **Response to Reviewer Uy67**
>
> We thank you for recognizing our work as interesting, novel and elegant! We would like to address your concerns point-by-point:
>
> 1. **Per-parameter compression.** Your concern about the correctness of "per-parameter compression" is addressed in Theorem 3 in the appendix (A.2 of the original submission). We realize that we had not referenced it in the main text, and we have now fixed that in our revision. Theorem 3 proves that compression of independent subsets of parameters is equivalent to the joint compression with larger bit-width in terms of the expectation of the test function $\zeta$, which is what we use in accounting. Hence, independent subset compression does not interfere with our proofs made for network-wise compression.
> 2. **End-to-end algorithm.** DP-REC can be embedded within the usual FL learning loop. In the appendix of our revision, we provide an extended algorithm which should enable a practitioner to extend their FL code-base with DP-REC. With respect to choosing the bit-width $b$ and $\sigma$ in practice, we can refer to the standard methodology for tuning hyper-parameters under DP-constraints (see, for example, [1, Appendix D]). Generally, one can tune $b$ according to the desired compression rates, while making sure that the number of bits is sufficient for meaningful privacy according to Theorem 1. $\sigma$ can be adjusted to capture the empirical gradient standard deviation and might be tuned on the server based on public, potentially out-of-domain data. That being said, we chose $b=7$ based on experiments in Figure 2 and chose $\sigma$ based on grid-search for individual data-sets and models.
> 3. **Increasing accuracy at the cost of communication.** Please see our general reply to commonly raised points.
>
> ---
> [1] Abadi, M., Chu, A., Goodfellow, I., McMahan, H.B., Mironov, I., Talwar, K. and Zhang, L. Deep learning with differential privacy. In *Proceedings of the 2016 ACM SIGSAC Conference on Computer and Communications Security*, pp. 308–318, 2016.

---

> > ### Comment · Reviewer_Uy67 · 2021-11-20
> > **Increasing my score accordingly**
> >
> > Thanks for clarifying, Theorem 3 completely addresses my major concern. I will increase my score accordingly.

---

> > > ### Author Response · Authors · 2021-11-20
> > > **Thank you**
> > >
> > > Thank you for revising your score. We are happy we managed to address your concern.

---

### Official Review · Reviewer_5TiV · 2021-11-03

**Correctness:** 3
**Technical Novelty And Significance:** 4
**Empirical Novelty And Significance:** 3
**Recommendation:** 6
**Confidence:** 3

**Main Review:**

I have a few questions that I hope the authors can help clarify.
- How is sensitivity controlled? It seems not quite clear to me the sensitivity control over \phi_s can transfer to \Delta_k*.
- It is a bit hard to interpret Table 1 and Figure 1 as both communication cost and accuracy are different. And it seems that aggressive compression will cause more accuracy drop. Since the communication-accuracy tradeoff of DP-REC can be tuned by K, or for different numbers of parameters/layers, can DP-REC match DP-FedAvg accuracy with less aggressive compression?
- Could the authors provide the clip norm and noise for both DP-FedAvg and DP-REC with some explanation? Appendix C.2 provides some, but I would appreciate an “apple-to-apple” comparison and some explanation on why from a first glance, the noise in DP-REC can be much smaller.
- Is it possible to provide more baselines? For example, DP-FedAvg with compression?
- The two versions of EMNIST (3.4K vs 3.5K) may have different numbers of labels, probably want to double check that the comparison is correct for table 1. Or even better, try to reproduce DDG with the same setting.


**Summary Of The Paper:**

This paper studies differentially private algorithms in federated learning, and proposes to take advantage of the randomness in Relative Entropy Coding (REC) to achieve good privacy-utility trade offs while significantly saving the communication costs. On four benchmark datasets (MNIST, FEMNIST, Shakespeare, Stack Overflow linear regression), under same privacy epsilon, the proposed method can save communication for ~(4000, 20, 800, 100)  times with accuracy loss ~(15, 6, 10, 1) compared to DP-FedAvg.




======= after rebuttal ============
I appreciate all the clarification and improvement on the draft. This paper is a borderline to me for the remaining concerns.

I want to be more specific here on "apples-to-apples" comparison. There are three things considered: differential privacy, communication and accuracy. It would be useful to show how this method can help achieve a reasonably good metric by tuning the other two metrics. Only working in the low accuracy regime sounds like a big limitation, and did not compare with other stronger compression method further weaken the claim (as also mentioned by other reviewers).

The authors mention secure aggregation in their rebuttal. IIUC, secure aggregation is future work in this draft. It is not very convincing.

That being said, The idea seems to be interesting, and I will raise the score from borderline reject to borderline accept.



**Summary Of The Review:**

It is a bit hard to justify the significance as it is not an apples to apples comparison as both communication and accuracy are different.

---

> ### Author Response · Authors · 2021-11-16
> **Response to Reviewer 5TiV**
>
> We thank you for reviewing our work and posing insightful questions. We aim to answer them one-by-one here.
>
> 1. **Control of sensitivity.** Please note that we do not need to control sensitivity of $\Delta_{k^*}$ because it is the output of our privacy mechanism and not of the function we want to privatize. If we think of it in terms of a standard DP-SGD, $\Delta_{k^*}$ is the noised gradient, which does not need to be bounded. Our goal is to limit the sensitivity of $f(w^{(t)}, D_s) = w^{(t)}_s - w^{(t)}$. This in turn places a bound on the R\'enyi divergence between distributions $\mathcal{N}(\hat{\phi}^{(t)}_s, \sigma^2 I)$ and $\mathcal{N}(0, \sigma^2 I)$. This leads to the bound on $\varepsilon$ following our proof of Theorem 1 (see Appendix A.1).
> 2. **Increasing accuracy at the cost of communication.** Please see our general reply to commonly raised points.
> 3. **Clipping norms and noise scale in DP-REC vs DP-FedAvg.** Although the standard deviation of noise is smaller in the examples the reviewer refers to, the ratio of noise to update norms is actually higher for DP-REC. This is because the noise is calibrated to individual updates rather than averages. Consider, for example, FEMNIST experiments: for $\varepsilon = 3$, the ratio of sensitivity to noise for DP-REC is $1.35$ per client, while for DP-FedAvg it is $\frac{\sqrt{100}}{1.85}$ per client (for batches of 100 clients).
> 4. **Additional baselines.** Combining DP-FedAvg (or differential privacy in general) with compression is not a straightforward task, which is why it motivates our paper. There are two options: (i) ensure DP, then compress the update; (ii) compress, then ensure DP. Each option has its challenges. First, consider (i). Adding noise at the client and then compressing the update before transmission, does not allow to calibrate noise to the aggregate and use tighter composition theorems (e.g., Rényi accountant), degrading the privacy guarantee. This is what essentially led to the development of hybrid methods such as cpSGD and DDGauss. As this direction was researched in the line of work leading up to DDGauss, we take their method as the best representation of such approach. Let us now consider (ii). Once updates are compressed by the client, we cannot add noise directly, because it would negate any compression. So the client needs to transmit the update first, using secure aggregation to protect against honest-but-curious server. We can compress updates using scalar quantization, but secure aggregation might add communication overhead countering some of the effects of compression (see [1]).  Even without the additional communication overhead of secure aggregation, we empirically observed that such a method can have both worse accuracy and worse communication efficiency than DP-REC and represents a rather weak baseline. We ran an experiment on our MNIST task and combined 8-bit scalar quantization of the client updates (in a way that satisfies the desired sensitivity) with DP-FedAvg with $\varepsilon=3$; there we observed that the global accuracy reached ${\sim}58\\%$ after 1k rounds, which is both smaller than the DP-REC result and significantly more expensive in terms of communication.
> Lastly, if we were to consider vector quantization to get ahead in terms of compression ratios, it would hinder application of secure aggregation, leaving us without any protection against honest-but-curious server (since adding noise is not possible either, as it would nullify compression). We have also included this discussion in the appendix of the revised paper.
> 5. **Versions of (F)EMNIST dataset.** Our version of FEMNIST (also available in the LEAF benchmark) has a larger number of clients (3.5k vs 3.4k). We mentioned this when explaining the experimental setup in the first paragraph of Section 4, but we noticed that the table caption showed incorrectly that we used the 3.4k client version and DDGauss the 3.5k one. We have fixed this in the revision. As for their practical differences, it can be seen in DP-FedAvg performance, which is the common baseline run on both datasets. For instance, for $\varepsilon = 3$, DP-FedAvg achieves around $74\\%$ for [2] (which uses 3.4k clients), and $74.2\\%$ on ours, whereas for $\varepsilon = 6$, DP-FedAvg achieves around $77.5\\%$ for [2] and $75.5\\%$ on ours. Although we admit that it's not a fully apples-to-apples comparison, we believe it does not affect the main insights from our evaluation.
>
> ---
> [1] Bonawitz, K., Ivanov, V., Kreuter, B., Marcedone, A., McMahan, H.B., Patel, S., Ramage, D., Segal, A. and Seth, K. Practical secure aggregation for privacy-preserving machine learning. In *Proceedings of the 2017 ACM SIGSAC Conference on Computer and Communications Security*, pp. 1175–1191, 2017.
>
> [2] Kairouz, P., Liu, Z. and Steinke, T. The distributed discrete gaussian mechanism for federated learning with secure aggregation. *arXiv preprint arXiv:2102.06387*, 2021.

---

> > ### Comment · Reviewer_5TiV · 2021-11-18
> > **More clarification**
> >
> > Thanks for the response. My major concern is still that it seems hard to do an apples-to-apples comparison.
> >
> > For DP-FedAvg + compression baseline. If we only compare central DP and clip after communication, potentially any compression method could work. Since DP-REC would communicate the random seeds, could the authors clarify how local/distributed DP is achieved, and how multiple clients are used in each round?

---

> > > ### Author Response · Authors · 2021-11-18
> > > **Response to Additional Questions by Reviewer 5TiV**
> > >
> > > Thank you for reading our reply and providing additional comments. Let us try to further clarify these points.
> > >
> > > 1. **Comparison to other methods.** From our understanding (please correct us if we misinterpret your point), an "apples-to-apples" comparison you are referring to would be to either equalize accuracy and then compare communication, or equalize communication and compare accuracy. The first one is actually done in the paper: the horizontal line on the plots, which reflects communication benefits, is calculated at the *same level of accuracy*. As for the second, equalizing communication, we could potentially force DDGauss or DP-FedAvg with scalar quantization to train in the same low-budget communication regime (e.g., at about 4 bits per parameter on FEMNIST), but it would be extremely detrimental to their performance (e.g., even for 12 bits per parameter DDGauss is not training reliably). We would essentially be comparing to methods that are either diverging or performing very poorly (please refer also to our comment on the performance of DP-FedAvg with 8 bits in the previous response). Therefore, we did not see a reason for such evaluation. Finally, if we bring privacy into equation, that becomes a 3-sided problem, and as we explained in the general response, it is not straightforward to equalize two of the three variables. However, given a better amplification (e.g., in the case of StackOverflow), we can aim for the same privacy budget and compare communication for a given accuracy, or vice-versa, which is essentially the result we report in the paper. For practically equivalent accuracy levels (which can be picked along the curves) and privacy, we demonstrate a consistent level of compression (about 100-fold) of DP-REC.
> > >
> > > 2. **Simpler compression baseline.** Unfortunately, the approach of compressing the updates first and then clipping on the server would have the same problems we highlighted. Namely, in order to protect against honest-but-curious server, we need to employ secure aggregation since updates are not obfuscated by the client. Scalar quantization is compatible with that; however, it leads to poor results in terms of both compression and accuracy. More advanced compression, on the other hand, is not necessarily compatible with secure aggregation, irrespective of where the clipping is done.
> > >
> > > 3. **Local guarantee.** There are in fact two random seeds chosen by the client, as we mention in Section 2.4 ("Shared random seed"). One is used to draw a "codebook", i.e. samples from the Gaussian distribution on which to encode the gradient. The other is used to draw from the categorical distribution $q_\pi$ formed on this codebook. Only the first seed is shared with the server, while the second remains secret. The $\delta$ we compute then reflects the "joint failure probability," when draws from both of these sources of randomness simultaneously are such that the mechanism fails to remain within the $\varepsilon$ bound on privacy loss. As such, publishing one of the seeds does not further affect this guarantee, and does not allow the server to reconstruct the original gradient (it only gets to see a "codeword").
> > >
> > > 4. **Multiple clients in a round.** Multiple clients in DP-REC are used in a similar way to DP-FedAvg. Let us give an example. Let's say the server samples 100 clients. They receive the current model, run their local gradient descent as usual, and then perform the client side of the REC compression (with predefined $C$ and $\sigma$, of course), all independently of each other. Then, after the server has received all the indices and seeds and has reconstructed the gradients, they are averaged and the model is updated. At the same time, since the sampling is done with replacement and noise is calibrated to the individual clipping thresholds ($C$), rather than the aggregate ($C/100$), we can account this whole round as if 100 clients were sampled sequentially one by one. This leads to better amplification.
> > >
> > > We hope our answers bring more clarity and we will be happy to address any further concerns.

---

> > > > ### Author Response · Authors · 2021-11-29
> > > > **Response to reviewer 5TiV**
> > > >
> > > > Dear reviewer,
> > > >
> > > > We hope to have addressed your last concern through our rebuttal. As the discussion period is ending soon, we would appreciate if you could acknowledge our rebuttal and let us know if you have any further questions. Thank you very much for your time and reviewing our work.

---

> > > > > ### Comment · Reviewer_5TiV · 2021-11-29
> > > > > **Still borderline due to remaining concerns**
> > > > >
> > > > > Thanks for the reminder. I appreciate all the clarification and improvement on the draft. This paper is a borderline to me for the remaining concerns.
> > > > >
> > > > >  I want to be more specific here on "apples-to-apples" comparison. There are three things considered: differential privacy, communication and accuracy. It would be useful to show how this method can help achieve a *reasonably good* metric by tuning the other two metrics. Only working in the low accuracy regime sounds like a  big limitation, and did not compare with other stronger compression method further weaken the claim (as also mentioned by other reviewers).
> > > > >
> > > > > The authors mention secure aggregation in their rebuttal. IIUC, secure aggregation is *future work* in this draft. It is not very convincing.
> > > > >
> > > > > That being said, The idea seems to be interesting, and I will raise the score from borderline reject to borderline accept.

---

### Author Response · Authors · 2021-11-16
**General response and commonly raised points**

First of all, we would like to thank all the reviewers for their time and valuable comments. We have written individual replies, where we try to address all of the raised concerns. In addition to that, we prepared and uploaded a revised version of the paper that includes extended discussions and a few extra results generated based on the reviews, clarifications and minor fixes. Finally, in this reply, we answer the question that we found common across several reviews.

A point mentioned by multiple reviewers (5TiV, Uy67, qyw3) concerns the accuracy drop of DP-REC and a potential trade-off between higher communication budget (higher K) and better accuracy.

Unfortunately, it is not simple to close this accuracy gap. The reason for observing the gap lies not in compression but rather in privacy. Figure 2 shows empirically that increasing bit-width has marginal returns, up until doing no compression at all. Any configuration in terms of higher bit-widths or more fine-grained vector quantization can be expected to perform between 7-bit per-tensor quantization and no-compression. Compression-only experiments in the newly added Appendix D.4 also corroborate this point, as the non-private compressed model achieves performance much closer to FedAvg. Expressing the divergence bound of discrete distributions (which we can't use directly, since they leak sensitive information) through continuous distributions leads to nearly double the amount of noise necessary for an equivalent guarantee compared to the regular continuous Gaussian mechanism. Thus, we would need to relax privacy to close the accuracy gap.

However, there is still a possible way to reduce the accuracy difference by employing stronger privacy amplification (e.g. adding a secure shuffler), which will counter the overhead of the bound. This effect is seen in StackOverflow dataset, where subsampling amplification is stronger due to a large number of users and the accuracy gap between DP-FedAvg and DP-REC is drastically smaller.

Besides that, we also noticed a small discrepancy when accounting our GB of communication for DP-REC in the case of FEMNIST. There, we used 20 bits for the random seeds whereas we used 32 bits for all the other experiments. We have since then corrected that, which lead to a small increase in the total communication from $12.3$GB to $14.2$GB. This does not change the overall conclusions.

---

### Decision · Program_Chairs · 2022-01-20

**Decision:**

Reject

**Comment:**

This submission describes an approach to compressing the communication in federated learning. The key idea is using a set of random samples from a prior distribution and then performing importance weighed sampling. The work performs an analysis of the privacy guarantees of this process and experimental evaluation.
The main issue with this work is the authors appear to be unaware that the basic problem they pose is solved in a more comprehensive and lossless way in a recent work https://arxiv.org/abs/2102.12099 (Feldman and Talwar, ICML 2021). That work shows that any differentially private randomizer can be compressed via a simpler algorithm that performs rejection sampling using a PRG. The algorithm does not loose privacy or utility (under standard cryptographic assumptions) while guaranteeing low communication. In contrast this work loses significantly in utility and provides opaque privacy guarantees.
This submission analyzes  a randomized that adds Gaussian distribution and, in particular, the compression technique in (Feldman and Talwar) applies to it. The technique proposed in this work is very similar in spirit (with prior distribution corresponding to reference distribution in the earlier work.
In light of the earlier work I do not think the contributions in this submission are sufficient for publication.